JCB Journal of Cell Biology

# Tepsin and AP4 mediate transport from the trans-Golgi to the plant-like vacuole in toxoplasma

Janessa Grech[1], Abhishek Prakash Shinde[5,6], Javier Periz[1], Mirko Singer[1,2], Simon Gras[1], Ignasi Forné[3], Andreas Klingl[4], Joel B. Dacks[6,7,8], Elena Jiménez-Ruiz[1], and Markus Meissner[1]

Apicomplexan parasites are obligate intracellular pathogens possessing unique organelles but lacking several components of the membrane trafficking machinery conserved in other eukaryotes. While some of these components have been lost during evolution, others remain undetectable by standard bioinformatics approaches. Using a conditional splitCas9 system in *Toxoplasma gondii*, we previously identified TGGT1_301410, a hypothetical gene conserved among apicomplexans, as a potential trafficking factor. Here, we show that TGGT1_301410 is a distant ortholog of *T. gondii* tepsin (*Tg*TEP), localized to the trans-Golgi and functioning as an accessory protein of the adaptor protein complex 4 (AP4). We demonstrate that AP4-*Tg*TEP is essential for the actin-dependent transport of vesicles to the plant-like vacuole (PLVAC) and Golgi organization. Notably, our findings reveal that, unlike in metazoans, the AP4 complex in *T. gondii* utilizes clathrin as a coat protein, a mechanism more closely aligned with that of plants. These results underscore a conserved yet functionally adapted vesicular transport system in Apicomplexa.

## Introduction

With >5,000 species, apicomplexan parasites are among the most devastating pathogens infecting humans and animals, causing substantial morbidity and mortality worldwide (Mathur et al., 2021).

As obligate intracellular parasites, apicomplexans must invade host cells, replicate within a parasitophorous vacuole, and finally exit the host cell to infect new cells (Blader et al., 2015). The adaptation to an intracellular lifestyle led to the evolution of unique cytoskeletal structures, signaling cascades, and organelles, such as micronemes and rhoptries, which are secreted during invasion (Cova et al., 2022), and the apicoplast, a non-photosynthetic organelle derived from red algae (McFadden and Yeh, 2017).

In addition, Apicomplexa are phylogenetically distant from their hosts, belonging to the superphylum of Alveolata that also includes dinoflagellates and Ciliophora. Although morphologically diverse, members of Alveolata share unique secretory organelles and an alveolar membrane system, called the inner membrane complex (IMC) in apicomplexans (Harding et al., 2016; Jimenez-Ruiz et al., 2016; Keeling et al., 2005).

Due to this phylogenetic distance, many conserved proteins are difficult to identify using standard bioinformatic approaches, as evidenced by the high frequency of genes annotated as "hypothetical" in the genomes of apicomplexans. Furthermore, certain pathways, like those for vesicular transport appear to be reduced, with apicomplexan parasites only containing 9–14 Rab-GTPases (Langsley et al., 2008), as compared with the inferred ancestral complement of 19–23 Rabs in the last eukaryotic common ancestor (Diekmann et al., 2011; Elias et al., 2012). Similar reductions are seen in other trafficking components, including lineage-specific losses of entire complexes (Klinger et al., 2024; Woo et al., 2015). This reduction is counteracted by the lineage-specific expansion of paralogs, such as ArlX proteins, that have been identified using a newly developed bioinformatic screening pipeline (Klinger et al., 2024).

In parallel, unique proteins have been implicated in endocytosis and recycling (Gras et al., 2019; McGovern et al., 2018; Spielmann et al., 2020), and endocytic structures, corresponding to the micropore (Nichols et al., 1994), have been identified to be required for uptake of exogenous material (Koreny et al., 2023; Schmidt et al., 2023; Wan et al., 2023).

Despite the central role of clathrin-mediated endocytosis (CME) in other eukaryotes, apicomplexan parasites appear to

[1]Experimental Parasitology, Department of Veterinary Sciences, Faculty of Veterinary Medicine, Ludwig-Maximilians-Universität, LMU, Munich, Germany; [2]Integrative Parasitology, Center for Infectious Diseases, Heidelberg University Medical Faculty, Heidelberg, Germany; [3]Zentrallabor für Proteinanalytik, Biomedical Center Munich, Ludwig-Maximilians-Universität, LMU, Munich, Germany; [4]Pflanzliche Entwicklungsbiologie, Biozentrum der Ludwig-Maximilians-Universität, Munich, Germany; [5]Department of Parasitology, Faculty of Science, Charles University, BIOCEV, Vestec, Czech Republic; [6]Division of Infectious Diseases, Department of Medicine and Department of Biological Sciences, University of Alberta, Edmonton, Canada; [7]Centre for LIfe's Origin and Evolution, Department of Genetics, Evolution, & Environment, University College, London, UK; [8]Institute of Parasitology, Biology Centre, Czech Academy of Sciences, České Budějovice (Budweis), Czech Republic.

Correspondence to Markus Meissner: markus.meissner@para.vetmed.uni-muenchen.de; Elena Jiménez-Ruiz: elena.jimenez@para.vetmed.uni-muenchen.de.

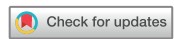

have largely dispensed with this pathway, as clathrin function is primarily restricted to postGolgi trafficking rather than endocytic vesicle formation (Pieperhoff et al., 2013). Instead, the uptake of extracellular material occurs through clathrin-independent mechanisms, such as micropore-mediated endocytosis, which subsequently feeds into the plant-like vacuolar compartment (PLVAC), a digestive organelle required for processing internalized material (Stasic et al., 2022). This deviation from the conventional CME pathway highlights a fundamental divergence in the evolution of membrane trafficking in these parasites, necessitating alternative regulatory mechanisms for endocytosis and lysosomal digestion. As a key component of this alternative pathway, the PLVAC acts as a digestive organelle containing multiple hydrolytic enzymes and a myriad of pumps and channels to regulate and maintain an acidic pH in this compartment (Stasic et al., 2022). The PLVAC not only participates in the digestion of host cell material (McGovern et al., 2018) but also regulates organelle turnover and recycling via autophagy (Di Cristina et al., 2017). It was proposed that the endosomal-like compartment (ELC), an intermediate secretory organelle, is not only responsible for the biogenesis of the secretory organelles micronemes, rhoptries, and dense granules but also of the PLVAC via Rab7, Mon1/Ccz1, and HOPS (Venugopal and Marion, 2018).

A similar knowledge gap exists regarding the role of the cytoskeleton in membrane trafficking in apicomplexans. While earlier studies focused on the role of microtubules in vesicular transport, recent findings highlight F-actin, in coordination with the motor protein myosin F (MyoF), as critical factors for vesicular transport (Carmeille et al., 2021; Heaslip et al., 2016; Periz et al., 2017; Stortz et al., 2019). Surprisingly, these parasites encode only a minimal set of actin-regulatory proteins in contrast to other eukaryotes (Das et al., 2021). We hypothesized that these parasites evolved a unique set of proteins that are (directly or indirectly) involved in F-actin regulation and F-actin–dependent vesicular transport. To identify novel, critical factors involved in F-actin dynamics, apicoplast inheritance, and/or host cell egress, we previously performed a phenotypic screen and obtained 16 candidates (Li et al., 2022). One, TGGT1_301410, showed altered F-actin patterns upon deletion and localized near the Golgi (Li et al., 2022), akin to Formin-2 (FRM2), a known actin regulator (Stortz et al., 2019), making it a priority target for further analysis.

TGGT1_301410 lacks a signal peptide or transmembrane domains but contains an ENTH domain, which is a structural domain that is often found in proteins involved in vesicular trafficking, such as epsins or tepsins (De Camilli et al., 2002). Here, we have performed in depth phylogenetic analysis and structural modelling approaches to demonstrate that TGGT1_301410 is a distant ortholog of tepsin that is conserved in apicomplexans. We demonstrate that TgTepsin (TgTEP) plays a critical role in the organization of the trans- but not cis-Golgi. Interestingly, deletion of TgTEP led to the accumulation of the PLVAC-resident cathepsin L (CPL) (Parussini et al., 2010) in the Golgi body. Proximity-labelling and co-immunoprecipitation (co-IP) experiments demonstrated that TgTEP interacts with the adaptor protein complex 4 (AP4) and, surprisingly, with

clathrin. Furthermore, the vesicles originated by TgTEP/AP4 are trafficked in MyoF- and actin-dependent manner, and its depletion results in a reduction of F-actin and MyoF dynamics. Finally, we reveal that although the tepsin–AP4 complex has been lost convergently across the tree of eukaryotes, the co-occurrence of tepsin and AP4 across eukaryotes is statistically significant, highlighting an ancient, conserved interaction.

## Results

### TgTEP and AP4 are well conserved across parasitic apicomplexans despite the phylogenetic distances with their host

In a previous study, using a phenotypic screen based on split-Cas9, we identified TGGT1_301410 as a candidate for regulation of F-actin dynamics and apicoplast inheritance (Li et al., 2022). Initial localization analysis demonstrated that this protein localizes near the Golgi apparatus of the parasite, where the actin-nucleator FRM2 is localized (Stortz et al., 2019).

TGGT1_301410 encodes a 1,033 amino acid protein, initially annotated as hypothetical in ToxoDB version 59 (Alvarez-Jarreta et al., 2024). It belongs to the ortholog group OG6_138217 and is highly conserved within alveolates, where it can be found as a single copy gene (Amos et al., 2022). TGGT1_301410 does not have a signal peptide or transmembrane domains and contains an ENTH domain from amino acids 1–150 (Fig. 1 A). Further analysis of available datasets suggests that TGGT1_301410 transcript levels are similar across different Toxoplasma gondii strains (GT1 and ME49) and life stages (tachyzoites and bradyzoites) (Behnke et al., 2008), with only a slight decrease during G1 phase (Behnke et al., 2010).

A previous phylogenetic analysis of proteins containing ENTH domains predicted that the ENTH domain of PF3D7_1459600, the ortholog of TGGT1_301410 in Plasmodium falciparum belongs to the tepsin family (Archuleta et al., 2017). However, unlike canonical tepsins, TGGT1_301410 lacks the internal VHS/ENTH-like (tVHS) domain. Given that tepsins are known accessory subunits of the AP4 adaptor complex, we sought to investigate whether TGGT1_301410 is a bona fide tepsin ortholog and whether AP4 is conserved across apicomplexans.

We used molecular evolutionary and structural modelling approaches to provide evidence of conservation for tepsin and AP4 in T. gondii and other apicomplexans. Comparative genomics for the homology searches of AP1, AP4 adaptor complex subunits, and their accessory components, epsin and tepsin, was conducted in 10 selected apicomplexan species (Fig. 1 B), with the AP subunits confirmed by phylogenetic analysis (Data S1). Every apicomplexan under survey possesses all the components for both the adaptor complexes. Epsin and tepsin homologs were identified in Gregarina niphandrodes, as confirmed by the relatively low BUSCO score for this dataset (55.7% of missing and 11.8% of fragmented), as listed in EukProt v3 (Table S1).

Phylogenetic analysis using all the identified epsin and tepsin candidate proteins was conducted using the respective human protein sequences and the previously established P. falciparum tepsin sequence (Archuleta et al., 2017) as reference. We observed distinct clades in the phylogeny of the identified

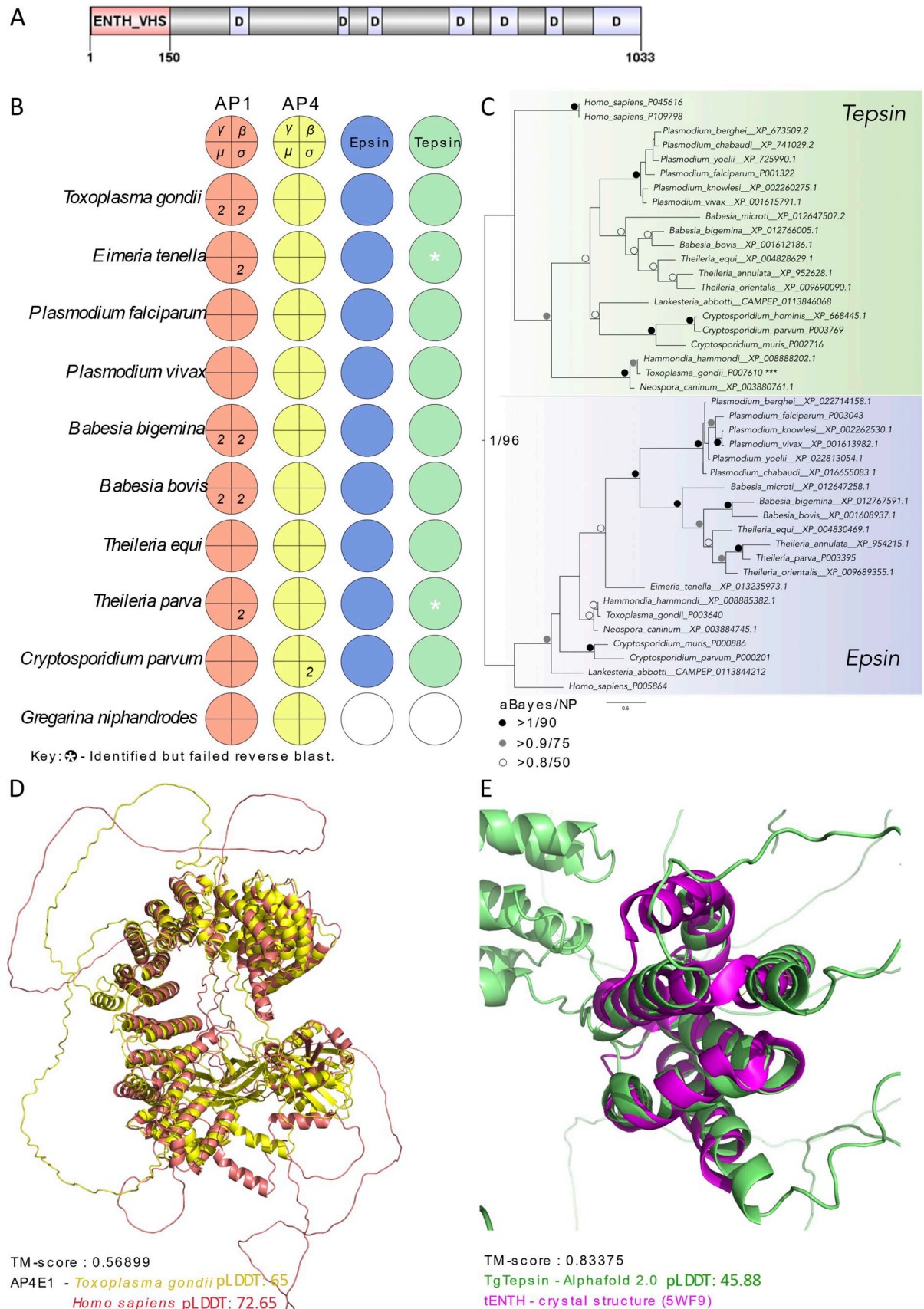

Figure 1. **Evolution and structural assessment of adaptor complexes and accessory components in Apicomplexa. (A)** Schematic of *Tg*TEP protein architecture showing an ENTH_VHS domain and disordered regions (marked "D"), with no additional annotated motifs. Generated using DOG 2.0 (Ren et al.,

2009). **(B)** Coulson plot representing the result of HMMer searches in 10 selected apicomplexans. Colored areas represent presence of protein homologs, white sections represent absence or unidentified, and numbers represent multiple paralogs identified. Respective proteins are annotated along with the binomial names of the species. **(C)** Phylogenetic analysis of epsin and tepsin using human protein sequences as references. aBayes represents Bayesian posterior probability calculations; NP represents nonparametric bootstrap support values. Clade in green represents tepsin; clade in blue represents epsin. **(D)** 3D superimposition of predicted AP4-ε structures of human (red) and *T. gondii* (yellow), TM-score, and pLDDT are provided. **(E)** 3D superimposition of the predicted *Tg*TEP (green) with the crystal of human tepsin ENTH domain (magenta), TM-score, and pLDDT are provided.

apicomplexan tepsin and epsin sequences (Fig. 1 C), supported with Bayesian posterior probability of 1.00 and 96% maximum likelihood nonparametric bootstrap support. This analysis provides evidence that *Tg*TEP (TGGT1_301410) and the human tepsin are distant yet conserved orthologs.

We also conducted structural assessment of *Tg*TEP and AP4-ε using human tepsin and AP4-ε as reference. Predicted structures of human and *T. gondii* AP4-ε were superimposed and structurally aligned using TM-align for visual and mathematical confirmation, respectively, of the protein folds. TM-align value of 0.57 for AP4-ε of both the distant species indicates that both these protein structures to possess the same structural fold (Fig. 1 D). Similarly, the TM-align value of 0.834 for the predicted structure of *Tg*TEP and crystal structure of human tepsin ENTH functional domain, despite the low pLDDT score for *Tg*TEP, indicates both the structures to possess the same folds for the conserved functional domain (Fig. 1 E).

Finally, we provide secondary structure alignments (Fig. S1 A), showing the conservation of identified functional residues in *Tg*TEP and other identified apicomplexan tepsins with the human tepsin ENTH domain responsible for interaction for membrane recruitment with AP4 complex. The essential residues of the ENTH domain include Arg10 and Glu55 responsible for ion pairing between alpha helices 1 and 3, both conserved in apicomplexans along with the Leu53 and Tyr56 responsible for hydrophobic interactions (Archuleta et al., 2017).

### Tepsin and AP4 are required for trans-Golgi organization and trafficking to the plant-like vacuole

To enable live imaging and localization analysis of *Tg*TEP, the gene was tagged with fluorescent or self-labelling tags (Fig. S1, B–D). All versions of endogenously tagged *Tg*TEP localized identically to the Golgi body, with some signal also observed throughout the cytoplasm, indicating a dynamic protein (Fig. 2, A–D and Fig. S1 D).

To define the location of *Tg*TEP at the Golgi and trans-Golgi network (TGN), co-staining with GRASP-RFP (Fig. 2, A and B), a marker for the cis-Golgi (Pfluger et al., 2005), and SortLR-Halo (Fig. 2, C and D), a marker for the trans-Golgi (Sloves et al., 2012), was performed. Confocal images (Fig. 2 C), orthogonal views (Fig. S1 D), and profile plots (Fig. 2 D) confirm that the protein is present in close proximity to the trans-Golgi but does not perfectly colocalize with SortLR (mean Pearson correlation coefficient between *Tg*TEP and SortLR was 0.546).

Further trans-Golgi markers, such as syntaxin-6 (Jackson et al., 2013), the dynamin-related protein B (DrpB; [Breinich et al., 2009]) an ELC marker proM2AP (Harper et al., 2006) and a PLVAC marker CPL (Miranda et al., 2010; Parussini et al., 2010), were used and established that *Tg*TEP is localized mainly

to the trans-Golgi (Fig. 2, E–L). Although all of these markers appeared to localize close to *Tg*TEP, no perfect colocalization could be observed, suggesting that it resides at a well-defined subcompartment of the trans-Golgi.

Lack of signal peptides/transmembrane regions suggests cytosolic Golgi association. To confirm this, we expressed a cytosolic single-chain nanobody fused to a Halo-tag that recognizes SYFP2 (GFP-Nb-Halo, Lee et al., 2014). This nanobody can only bind to *Tg*TEP-SYFP2 in case it is localized on the cytosolic side of the vesicular membrane (Fig. 3 A). Indeed, *Tg*TEP-YFP localized efficiently with GFP-Nb-Halo at the Golgi, demonstrating that *Tg*TEP is present within the cytoplasm and potentially bound to the Golgi via interactions with other proteins (Fig. 3, B and C). Control strains expressing YFP-tagged SAG1 (extracellular), FRM2 (cytosolic), and *Tg*TEP-mCherry (absence of YFP) validated specificity of this assay (Fig. 3, D–G).

Finally, brefeldin A (BFA) was used to disrupt the Golgi body (Helms and Rothman, 1992; Sciaky et al., 1997), which led to the redistribution of both *Tg*TEP and SortLR (Fig. 3 H), confirming their association with the Golgi.

To validate the phenotype obtained in the sCas9 screen (Li et al., 2022), we generated a conditional null mutant for *Tg*TEP using the DiCre system (Andenmatten et al., 2013). Therefore, *Tg*TEP was flanked with loxP sequences in RH DiCreΔku80Δhxgprt (Hunt et al., 2019) (Fig. S1 B). Induction with 50 nM rapamycin allowed efficient excision of the gene (Fig. S1 E) in ~90% of the parasite population with protein levels of *Tg*TEP being undetectable 48 h after induction (Fig. S1 F). Plaque assays confirmed the essentiality of the protein of interest, with no plaques being visible 7 days after induction of the knockout (Fig. 3 I). Analysis of the lytic cycle demonstrated that parasites devoid of *Tg*TEP are able to invade the host cell (Fig. S1 G) but subsequently fail to efficiently replicate and egress from the host cell (Fig. 3, J and K).

Knockout had no impact on cis-Golgi (GRASP-RFP) even after extended induction times of 72 h, but electron microscopy showed large electron-lucent vesicles and disrupted Golgi stacks (Fig. 4, A and B).

This observation aligns with our analysis of several trans-Golgi markers, including SortLR, DrpB, and syntaxin-6, which exhibited substantial vesiculation 48 h after *Tg*TEP depletion (Fig. 4, C–G). In contrast, the ELCs, located near the Golgi and visualized using the marker proM2AP (Harper et al., 2006), remained unaffected (Fig. 4 H). Similarly, no effect was observed on other organelles, such as micronemes, rhoptries, apicoplast and IMC (Fig. S2, A–E), even after extended induction times of 96 h (Fig. S2, A–G).

One notable exception was the mitochondrion, which fragmented at later stages, possibly due to indirect effects like

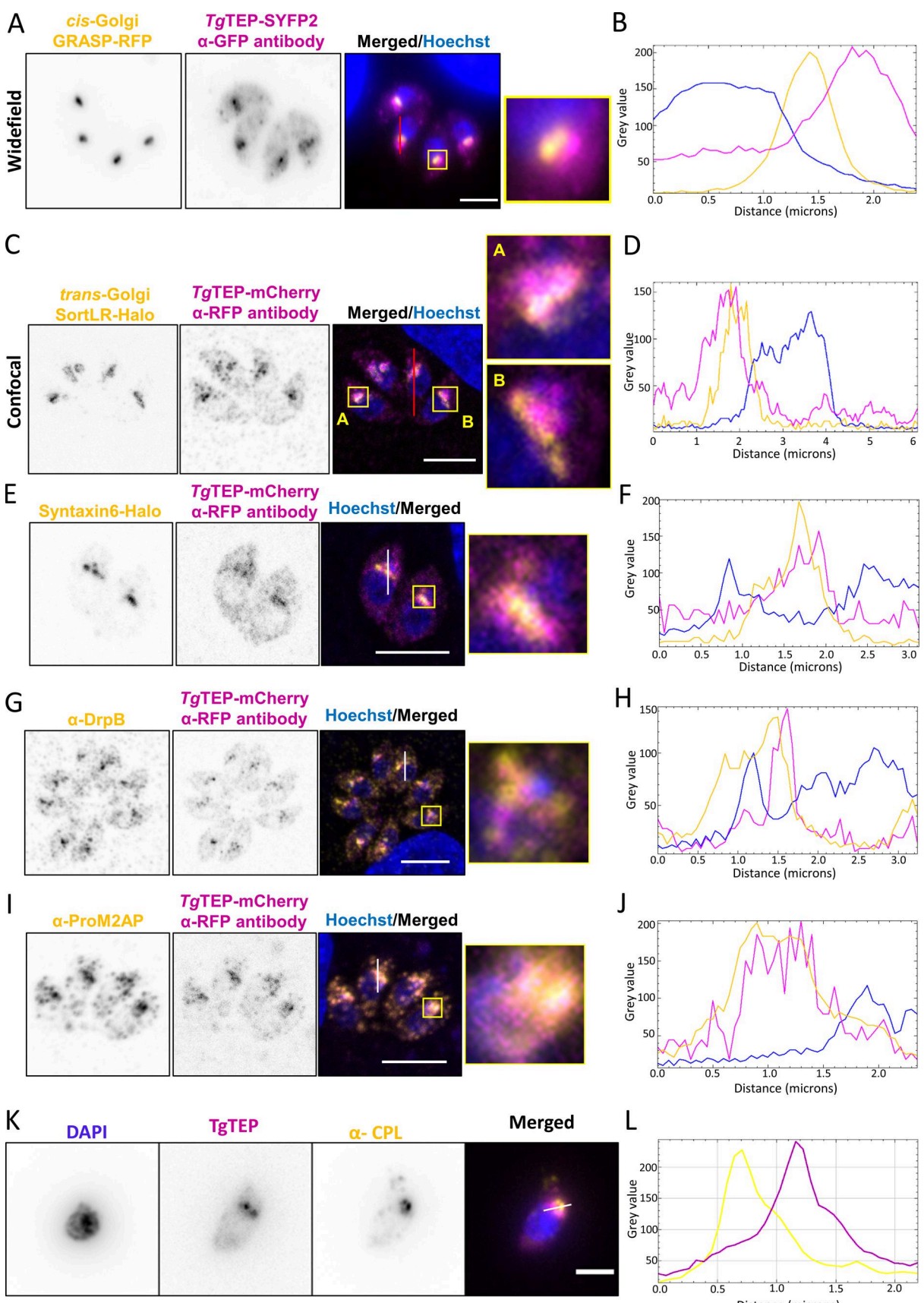

Figure 2.   **TgTEP localizes close to the trans-Golgi.** Colocalization analysis of *Tg*TEP (in magenta) with various markers (in yellow) done in triplicate with a minimum of 100 parasites observed per replicate. Profile plots were taken across the respective lines. **(A and B)** Parasites expressing GRASP-RFP (a cis-Golgi

marker) show no clear overlap with *Tg*TEP. **(C and D)** Parasites expressing SortLR-Halo (a marker for the trans-Golgi) show a partial overlap with *Tg*TEP. **(E–J)** Similarly, colocalization analysis with other postGolgi markers, such as parasites expressing syntaxin-6-Halo (E and F), stained with α-DrpB (G and H) or α-ProM2AP (I and J) show only a partial overlap with *Tg*TEP. **(K and L)** Colocalization analysis on extracellular parasites between *Tg*TEP and the PLVAC marker CPL shows no significant overlap. Scale bars: 5 μm (2 μm for CPL). Prior to obtaining the profile plots, images were converted to 8 bit for intensity normalization.

---

nutrient starvation (Fig. S2, F and G). While we cannot exclude that *Tg*TEP plays a direct role in mitochondrial division and/or segregation, based on its location and the strong Golgi-related phenotype, which occurs prior to mitochondrial fragmentation, we speculate that this defect is indirect. Similar phenotypes resulting in mitochondrial fragmentations have been described upon disruption or depletion of other Golgi-located proteins, such as clathrin (Pieperhoff et al., 2013) or factors involved in autophagy (Besteiro et al., 2011; Nguyen et al., 2017; Smith et al., 2021, *Preprint*).

To validate interaction between *Tg*TEP and AP4, we tagged AP4ε with Halo. Colocalization with *Tg*TEP-mCherry confirmed interaction, which was lost upon *Tg*TEP depletion (Fig. 5 A). Reciprocal co-IP experiments robustly confirmed the physical association between *Tg*TEP and AP4ε. co-IP was performed in a strain containing both *Tg*TEP and *Tg*AP4ε tagged with YFP and 3xHA tags, respectively (Fig. 5 B). Furthermore, mass spectrometry analysis of the immunoprecipitates revealed consistent enrichment of all four AP4 subunits (ε, β4, μ4, and σ4), as well as clathrin heavy and light chains, as well as a weaker interaction with DrpB (Fig. 5, C and D), which is localized at the trans-Golgi (Breinich et al., 2009). The significant enrichment of clathrin in both pull-downs suggests that AP4-dependent transport might be similar to plants, where transport to the vacuole depends on AP4 and clathrin (Dahhan et al., 2022; Fuji et al., 2016).

CRT, a PLVAC transporter, was also enriched, suggesting vesicles trafficked by *Tg*TEP-AP4 are PLVAC bound (Warring et al., 2014).

### *Tg*TEP and AP4 direct clathrin-dependent trafficking to the PLVAC

*T. gondii* shares many features with plants, including the organization of the endomembrane system, which contains a PLVAC (Stasic et al., 2022). This compartment was shown to be involved in the digestion of host cytosolic proteins and is, therefore, linked between the secretory and endocytic pathway (McGovern et al., 2018). While the current model predicts that proteins destined for the PLVAC traffic through ELCs (Stasic et al., 2022), we hypothesized that the situation might be similar as observed for plants, where the vacuole is formed by distinct but interdependent trafficking pathways with the TGN acting as central sorting station (Law et al., 2022).

To test the integrity and formation of the PLVAC, we used antibodies against CPL, a marker for this compartment (Miranda et al., 2010; Parussini et al., 2010). Since the PLVAC is mainly formed in extracellular parasites, we analyzed the distribution of CPL in intra- and extracellular parasites in dependence of *Tg*TEP expression. Upon excision of *Tg*TEP, CPL accumulates as soon as 24 h and peaks at around 48 h after induction (Fig. 5 E and Fig. S2 H). This phenotype was similarly observed in extracellular parasites (Fig. S2 I). To assess if CPL is blocked within

the TGN, we colocalized CPL with the TGN marker SortLR that was endogenously Halo-tagged. Indeed, CPL and SortLR colocalized and accumulated within the fragmented Golgi (Fig. 5 F). To extend this analysis, we generated a conditional null mutant for TgAP4ε using the DiCre system (Andenmatten et al., 2013). TgAP4ε knockout replicated this phenotype (Fig. 6, A–C), with CPL buildup preceding Golgi disruption (Fig. 6, C and D).

Together, these data demonstrate that *Tg*TEP and TgAP4ε are crucial for the specific trafficking to the PLVAC. The sequential events observed following the depletion of *Tg*TEP and TgAP4ε reveal that the disruption of trafficking to the PLVAC precedes Golgi collapse, which is subsequently followed by mitochondrial fragmentation (Fig. 6 D). The mitochondrial phenotype is most likely a result of a starvation response in the parasite, as observed previously (Ghosh et al., 2012).

### *Tg*TEP vesicles rely on actin and MyoF for intracellular transport

To identify transient trafficking factors interacting with *Tg*TEP and AP4 during vesicular transport, we employed TurboID, an enhanced biotinylation-based proximity labelling technique (Cho et al., 2020). Its high reactivity enables the detection of both stable and transient interactors with temporal resolution. TurboID efficiency was validated by immunofluorescence using streptavidin-coupled antibodies: without biotin, only the apicoplast was labelled, while 30-min biotinylation primarily marked the Golgi. After 6 h, labelling extended to a cloud around the Golgi and distinct foci at the parasite periphery and basal end (Fig. 6 E).

Comparative enrichment against TurboID-negative controls revealed multiple candidate interactors (Fig. S3, A–D), including AP4 subunits and clathrin—consistent with co-IP results (Fig. 5, C and D). All proteins detected at 30 min were also enriched at 6 h (Fig. 6 F). To refine candidates, the 30-min dataset was filtered by excluding proteins with a phenotypic score ≥ minus 1 (Sidik et al., 2016), those spatially distant from *Tg*TEP (based on hyperLOPIT; Barylyuk et al., 2020), and proteins unlikely to anchor *Tg*TEP, such as IMC or ribosomal proteins. The resulting candidates are listed in Tables S6, S7, S8, and S9.

Interestingly, and consistent with its identification in a forward genetic screen for actin-regulating proteins (Li et al., 2022), several proteins linked to actin functions, such as FRM2, MyoF, and profilin, were identified (Fig. 6 F and Fig. S3 D). Furthermore, F-actin, FRM2, and MyoF have been demonstrated to be required for vesicular transport and organization of the Golgi, and phenotypes are also seen upon depletion of *Tg*TEP and TgAP4ε. However, while depletion of F-actin, FRM2, and MyoF affect multiple trafficking pathways, implicating their involvement in different transport processes, such as dense granule motility, recycling of maternal material, or Golgi-organization (Carmeille et al., 2021; Das et al., 2021; Heaslip

Figure 3. **_Tg_TEP localizes in the cytoplasm and is essential for parasite survival. (A)** Scheme of strategy to determine the localization of _Tg_TEP. A GFP-nanobody fused to a Halo-tag was stably expressed within the parasites by replacing the UPRT locus. This nanobody has no target sequence and therefore

localizes within the cytoplasm. In cases where no GFP, YFP, or SYFP2 are accessible within the cytoplasm, no colocalization with the nanobody occurs, and the nanobody signal remains diffuse within the cytoplasm (left panel). In cases where GFP, YFP, or SYFP2 are present within the cytoplasm, the nanobody binds to the fluorescent tag, resulting in colocalization (right panel). **(B and C)** *Tg*TEP-SYFP2 colocalizes with the cytosolic nanobody. **(D–G)** FRM-2-SYFP2 (positive control), SAG1-YFP, and *Tg*TEP-mCherry (negative controls) confirm selective binding of the GFP-nanobody. **(H)** BFA disrupts Golgi, redistributing both *Tg*TEP and SortLR in all cases (100%). All scale bars are 5 µm. **(I)** 7-day plaque assays were done with LoxP-*Tg*TEP parasites in the presence of 50 nM rapamycin or vehicle control (DMSO). Knockout (KO) mutants (+rapamycin) did not form plaques in the host cell monolayer. Scale bars are 1.5 mm. **(J)** Replication assays were performed after the induction of parasites with 50 nM rapamycin or DMSO for a period of 48 h. The number of knockout parasites per vacuole was significantly lower than that of WT parasites. **(K)** Egress assays were done in the presence of 50 nM rapamycin or DMSO (–/+ Rapa). Egress was either allowed to occur naturally or was induced using calcium ionophore A23187 (–/+ CI). The percentage of successfully egressed parasites was significantly lower for the knockout mutants. All assays were done thrice, with a minimum of 100 parasites/vacuoles quantified per condition per replicate. All data are plotted as mean ± SD. One-way ANOVA with Tukey's multiple comparison test were performed. Color-coded P values in J represent the vacuoles compared. P values are represented as follows: ns ≥ 0.05; * = 0.01 - 0.05; ** = 0.001 – 0.01; *** = 0.0001 – 0.001; **** ≤0.0001.

---

et al., 2016), *Tg*TEP, and AP4ε appear to specifically mediate transport from the trans-Golgi to the PLVAC, indicating a more targeted functional role rather than broad actin regulation.

To investigate actin dynamics, we inserted the expression plasmid for actin chromobody-emerald (Periz et al., 2017) into the parasite strains where *Tg*TEP was already tagged and floxed (Fig. 7 A). Live imaging confirmed that *Tg*TEP vesicles move along actin filaments. These vesicles were not only observed to move along actin filaments around the Golgi region (Fig. 7 B and Video 1) but also along the periphery of the parasites (Fig. 7 C). Deletion of *Tg*TEP resulted in the disappearance of actin filaments at the Golgi region (Fig. 7 A), similar to what can be observed in the case of FRM2 depletion (Stortz et al., 2019). Remaining filaments appear to be slightly shorter and more concentrated toward the basal ends of the parasites (Fig. 7 A).

To further investigate the effect of the abrogation of *Tg*TEP on actin kinetics, we implement kymograph analysis to quantify actin chromobody changes in *Tg*TEP-cKO and WT parasite lines (Fig. 7, D and E). We found that representative kymotracks extracted from the remaining F-actin network did not have significant differences with those found in WT cells (Fig. 7 F). This argues for *Tg*TEP not having a direct role in controlling the polymerization of actin in the parasite cytoplasm, with smaller but stable F-actin network in *Tg*TEP parasites maintaining the same flow kinetics as WT parasites.

Next, we tested for the distribution of MyoF and FRM2 in the absence and presence of *Tg*TEP (Fig. 8). MyoF became punctate, immobile, and distributed throughout the parasite (Fig. 8, A and B; and Video 2). The number of vacuoles exhibiting this phenotype was seen to be significantly higher than WT 48 h after induction (Fig. 8 B).

In contrast, FRM2 remained unaffected and localized apically to the nucleus in *Tg*TEP-depleted parasites (Fig. 8 C) and vice versa (Fig. 8 D). To better understand the effect of *Tg*TEP on the trafficking on the actin network, we measured the relocation and displacement of MyoF in *Tg*TEP knockout and of *Tg*TEP in the FRM2 knockout cell lines (Fig. 8, E and F). We estimated two parameters, displacement and changes in particle speed on the MyoF and *Tg*TEP proteins; the results show how depletion of *Tg*TEP causes changes in displacement and dynamics in MyoF (Fig. 8 E and Video 2). However, the absence of FRM2 only slightly alters the displacement of *Tg*TEP (Fig. 8 F and Video 3).

In summary, the results obtained from measuring F-actin flow in the *Tg*TEP knockout support the notion that, while

*Tg*TEP may not be directly involved in supporting actin polymerization by regulating FRM2, it could play a role in regulating the membrane scaffold required for the association of FRM2 with F-actin in the cytoplasmic endomembrane and the incorporation of MyoF into the actin network (Fig. 8 G). These findings also suggest that *Tg*TEP-positive vesicles are transported in an F-actin–dependent manner, potentially involving a direct association between *Tg*TEP and the motor protein MyoF to facilitate transport. The observed displacement of *Tg*TEP upon the knockout of FRM2 (Fig. 8 F) aligns with this observation. These results suggest that the MyoF population, which normally interacts with *Tg*TEP, forms these clusters in the absence of *Tg*TEP. However, this effect does not appear to be essential for other MyoF-mediated transport pathways, such as dense granule motility.

### *Tg*TEP might transiently interact with the micropore but is not required for endocytosis

To identify potential transient interactors, the data obtained from the 6-h TurboID experiment were normalized to the 30-min experiment and analyzed (Fig. 6 F and Fig. S3 C), using the same selection criteria as before. Of the two proteins seen to be the most enriched, one was seen to be nonessential (TGGT1_229930), while the other protein was profilin. Additional enriched proteins included K13, Eps15L, and CGAR, all recently identified as components of the *T. gondii* micropore, a site of endocytosis (Koreny et al., 2023; Wan et al., 2023). Although *Tg*TEP is critical for trafficking to the PLVAC—a compartment that receives both endocytosed material and vesicles originating from the micropore (Stasic et al., 2022)—the minimal enrichment of micropore-associated proteins in pull-down assays suggests *Tg*TEP-positive vesicles may only transiently interact with the micropore, or alternatively, that *Tg*TEP is not directly involved in endocytosis.

We therefore performed endocytosis assays and analyzed the fate of SAG1-Halo labelled with a membrane non-permeable dye as previously described (Koreny et al., 2023). No significant changes were observed in the percentage of vacuoles with visible endocytic events, confirming that *Tg*TEP plays no direct role in endocytosis (Fig. S3, E and F). Interestingly, an increased proportion of the vacuoles exhibited enlarged endocytic vesicles in absence of *Tg*TEP (Fig. S3 G), suggesting a potential accumulation of vesicles destined to the PLVAC, which is disrupted in absence of *Tg*TEP. This supports the idea that *Tg*TEP vesicles

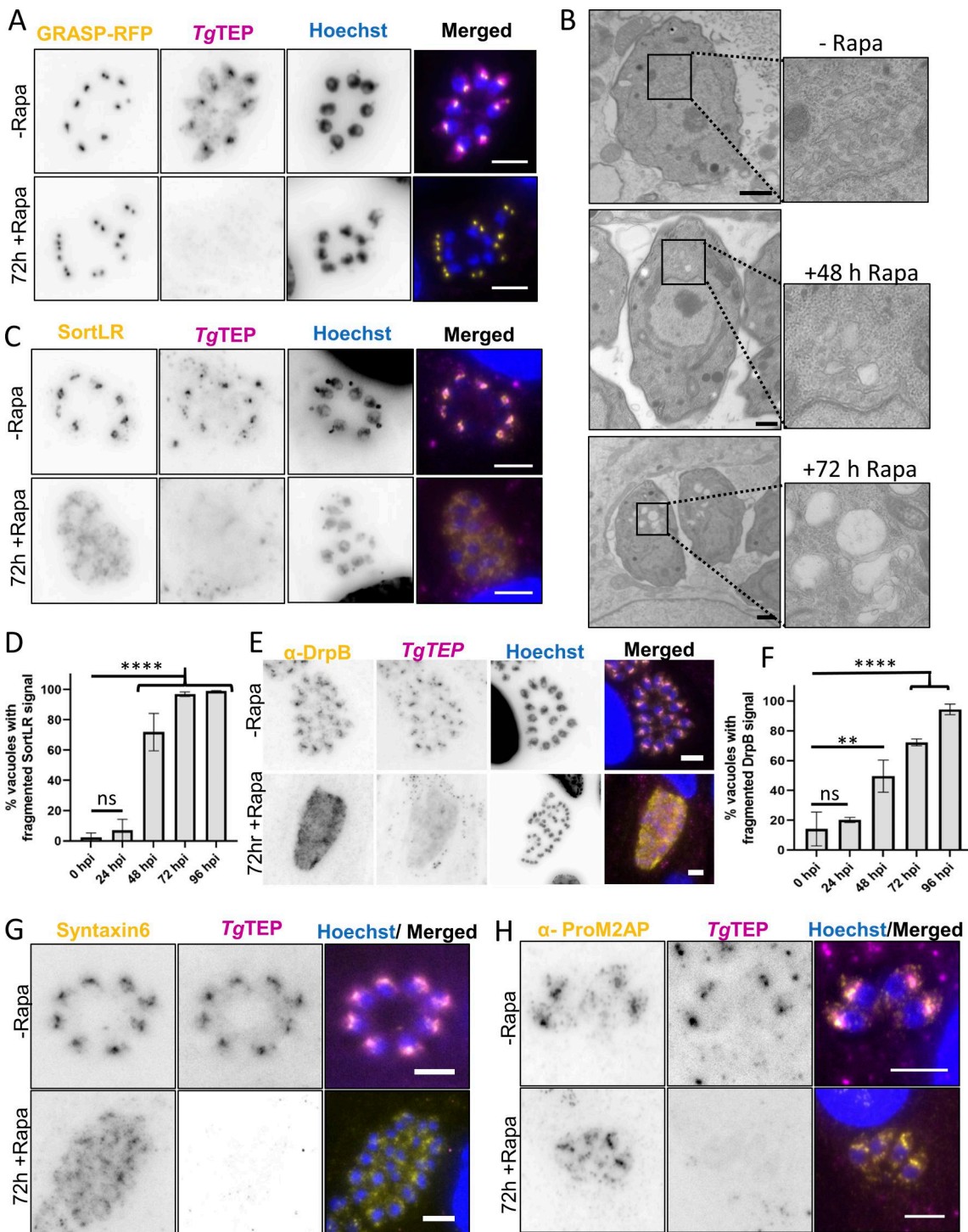

Figure 4.  **Knockout of *Tg*TEP results in trans-Golgi fragmentation. (A)** The cis-Golgi marked with GRASP-RFP (in yellow) is unaffected upon knockout of *Tg*TEP (shown in magenta). This was observed in 100% of cases. **(B)** TEM images demonstrate that in non-induced parasites (-Rapa), Golgi stacks are organized adjacent to the nucleus. At 48- and 72-h after treatment with 50 nM rapamycin (+Rapa), *Tg*TEP knockout results in the appearance of large electron-lucent vesicular structures and disruption of Golgi integrity. Scale bars: 1 μm. **(C)** Upon knockout of *Tg*TEP (in magenta), the trans-Golgi, labelled by endogenously tagged SortLR with Halo-tag (in yellow), was seen to fragment. **(D)** Quantification confirms significantly increased trans-Golgi fragmentation at 48 h post-induction (hpi) with rapamycin in *Tg*TEP-KO compared with controls. **(E)** The compartment marked by DrpB (yellow) is seen to fragment upon *Tg*TEP (in magenta) knockout. **(F)** Quantifications showed that in knockout parasites, the fragmentation of the post-Golgi compartment marked by DrpB was significantly higher than that in WT parasites after 48 hpi with rapamycin. **(G)** Syntaxin-6 distribution is similarly disrupted following *Tg*TEP loss. **(H)** Knockout of *Tg*TEP (in magenta) was not seen to affect the ELC labelled with anti-ProM2AP antibodies (in yellow). All immunofluorescence assays were done three times, with a minimum of 100 parasite vacuoles quantified per condition per replicate. Data are presented as mean ± SD. One-way ANOVA followed by Tukey's multiple comparison test were done, with P values being represented as follows: ns ≥ 0.05; ** = 0.001–0.01; **** ≤0.0001. All scale bars are 5 μm. KO, knockout.

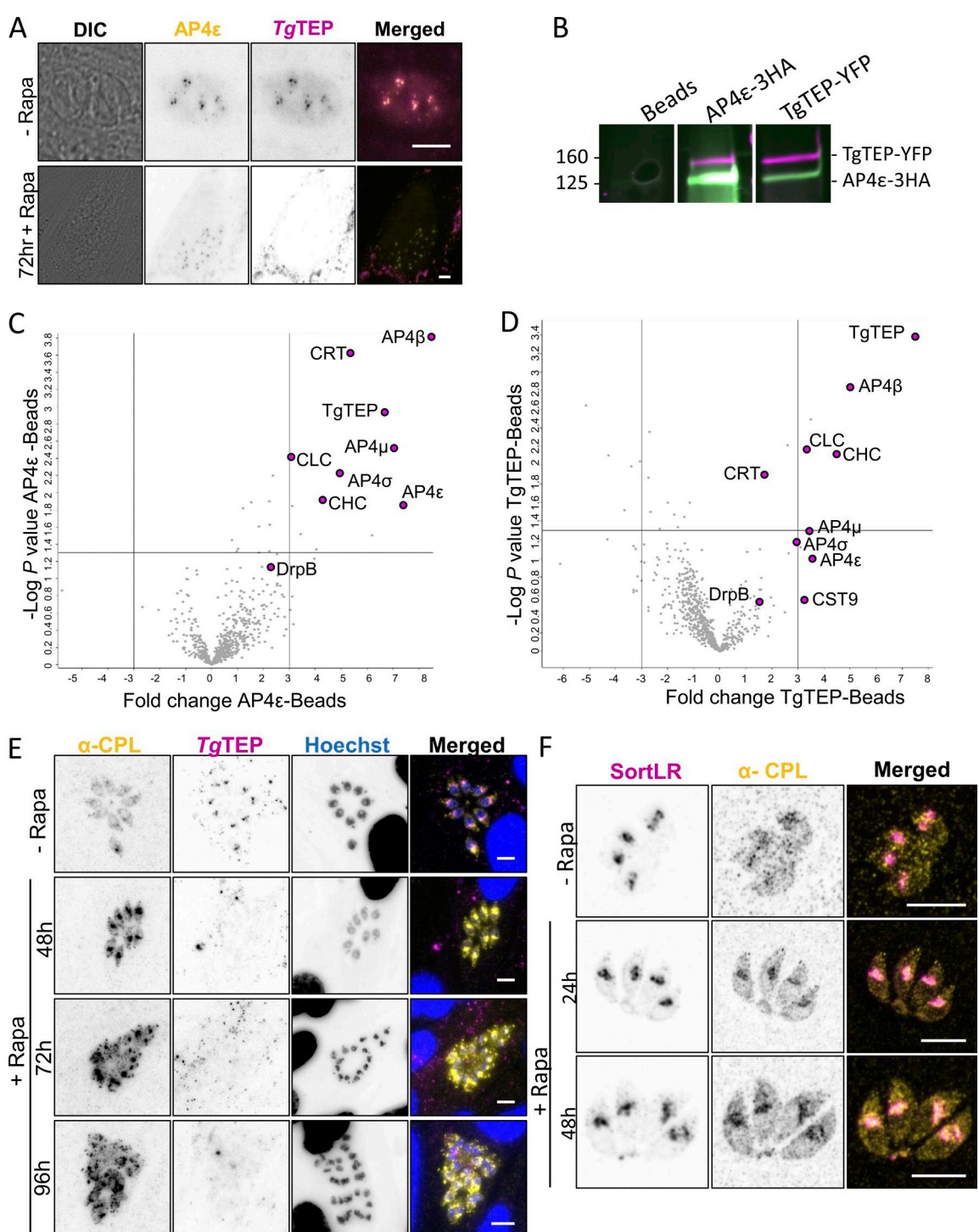

Figure 5. **TgTEP and TgAP4 interact in clathrin-mediated transport to the PLVAC. (A)** Colocalization experiments demonstrate that Halo-tagged AP-4ε (in yellow) colocalizes with mCherry-tagged TgTEP (in magenta). AP-4ε localization at the Golgi disappears in absence of TgTEP. All scale bars are 5 µm. **(B)** Western blot analysis of reciprocal co-IP assays confirms the physical interaction between TgTEP–GFP and AP4ε–HA. Beads alone, anti-GFP, and anti-HA–conjugated pull-downs are shown. Full blots are provided in Source Data. **(C and D)** Mass spectrometry of co-IP elutes reveals a significant enrichment of AP4 complex subunits and clathrin in both TgTEP and AP4ε pull-downs, displayed as volcano plots. Notably, CRT, a PLVAC transporter, is also enriched, supporting a role for this complex in PLVAC-directed trafficking. **(E)** CPL is found in small cytoplasmic vesicles in intracellular parasites and typically shows a diffuse localization. Depletion of TgTEP (in magenta) resulted in an accumulation of CPL (in yellow). This accumulation was seen as early as 48 h after induction, and fragmentation occurred at 72 h after induction of TgTEP knockout. Scale bars are 5 µm. **(F)** Colocalization of CPL (in yellow) with SortLR-Halo (in magenta) upon deletion of TgTEP demonstrates that CPL accumulation occurs in the trans-Golgi prior to its fragmentation. Scale bars are 5 µm. Source data are available for this figure: SourceData F5.

Figure 6. **_Tg_TEP and _Tg_AP4ε are essential for PLVAC trafficking and parasite survival. (A)** Plaque assay of WT and loxP-AP4ε-HA parasites shows a drastic diminution of plaques in parasites induced with 50 nM of rapamycin. **(B)** Quantification of plaque area. One-way ANOVA with Tukey's multiple comparison test was done. The P values are represented as follows: ns ≥ 0.05; * = 0.01; **** ≤0.0001. **(C)** Immunofluorescence imaging shows CPL (magenta), a PLVAC marker, accumulates in AP4ε-KO parasites (AP4ε shown in green) from 48 h after induction, mirroring the phenotype observed in _Tg_TEP-KO. Scale bar: 5 μm. **(D)** Temporal distribution of phenotype appearances in _Tg_TEP knockout parasites showing that disruption of CPL trafficking precedes Golgi fragmentation. Mitochondria collapse occurred significantly later. **(E)** _Tg_TEP was tagged with TurboID at the C terminus to carry out proximity-based biotinylation and find

interaction partners via the addition of 150 μM biotin. Immunofluorescence assays using fluorescently conjugated streptavidin show the localization of these biotinylated proteins (in magenta). Biotinylation for different lengths of time show different intensities and localizations of biotinylated proteins. Naturally occurring biotinylated proteins within the apicoplast are also labelled with the fluorescently conjugated streptavidin. The apicoplast was co-labelled with antibodies against G2-Trx (in yellow). All images are normalized. Scale bars are 5 μm. **(F)** All proteins identified at the 30-min time point were also identified at the 6-h time point. Proteins of high interest identified are listed and included those typically associated with the Golgi and postGolgi compartments (in blue), actin-binding proteins (in pink), proteins associated with parasite's endocytic micropore (in green), and an uncharacterized AP-4 subunit (in orange). KO, knockout.

---

may transiently fuse with endocytic vesicles at the TGN, with proximity labelling of micropore components occurring during this fusion event.

**Evolution of tepsin across eukaryotes is tightly co-related to the evolution of AP4 adaptor complex**

Having established details about the possible functions of tepsin in the *T. gondii* model system, particularly the critical interaction with the AP4 complex, we wanted to widen the scope and place this work in a pan-eukaryotic context. To do this, we conducted a comparative genomic survey (supported by phylogenetic confirmation of orthology [Data S1]) for the presence of AP4 subunits, epsin, and tepsin across 53 pan-eukaryotic species. The presence of AP1 subunits in those same 53 species was taken as a positive control, since the AP1 complex has never been reported as absent from any eukaryotic genome and is embryonically lethal in mouse models (Zizioli et al., 1999). In addition to further robustly confirming the orthology of *Tg*TEP via a pan-eukaryotic dataset, our analysis identified a pattern of AP4–tepsin co-occurrence, as per our speculation based on previous studies showing depletion of AP4 resulting in the loss of tepsin (Archuleta et al., 2017; Borner et al., 2012; Frazier et al., 2016) (Fig. 9).

In all the members of Telonemia, Stramenopiles, Alveolates, and Rhizaria supergroup, we observed the presence of AP4 and its accessory component, tepsin, alongside the presence of AP1 and its accessory component, epsin. We included three apicomplexan species: *P. falciparum*, *T. gondii*, and *Cryptosporidium parvum* for this broad analysis. All three apicomplexan species analyzed here showed the presence of AP1, AP4, and their respective accessory components. While epsin is seen to be expanded in Ciliates: *Tetrahymena thermophila* and *Paramecium tetraurelia* and Rhizarians: *Bigelowiella natans* and *Brevimastigomonas motovehiculus*, tepsin is present but not expanded in this supergroup except for *Thalassiosira pseudonana*. Similarly, AP1, epsin, AP4, and tepsin are present in all the species of supergroups Haptista, CRuMs, and Malawimonadida (Fig. 9 A).

We observed confirmed loss of tepsin in species with missing AP4 across eukaryotes. This can be seen in supergroups, plants (Archaeplastida): *Cyanidioschyzon merolae*, *Gloeochaete wittrockiana*, *Porphyra purpurea*, and *Cyanophora paradoxa*; Amorphea: yeast and *Drosophila melanogaster*; Discoba: *Leishmania major*; and Fornicates of Metamonada group such as *Giardia intestinalis* and Carpediemonas-like organisms. Additionally, we also observed the absence of tepsin despite the presence of AP4 in three pan-eukaryotic species; *Guillardia theta*, *Entamoeba histolytica*, and *Monocercomonoides exilis* (Fig. 9 A). This pattern of coevolution or co-occurrence was confirmed statistically with an independence chi-square test for all the surveyed species, where the two variables are AP4 and tepsin. The expected frequencies and chi-square points for the calculation of chi-square value ($\chi^2 = 38.67$) and significant P value were lower than 0.05 (Fig. 9 B). Out of the 53 species, 11 species have both lost AP4 and tepsin, while 39 species show the co-occurrence of AP4 and tepsin (Fig. 9 C).

## Discussion

Tepsins are widely conserved in eukaryotes and usually, in addition to a C-terminal ENTH domain, possess a second internal folded module, the VHS/ENTH-like domain (Archuleta et al., 2017). In contrast, *Tg*TEP (TGGT1_301410), identified as an interesting candidate in a phenotypic screen designed to identify proteins associated with changes in actin dynamics (Li et al., 2022), only contains a C-terminal ENTH domain. Using BlastP searches, no clear ortholog could be identified in eukaryotes outside of the apicomplexan phylum, but based on structural prediction and comparison (Foldseek; [van Kempen et al., 2023]), the closest putative ortholog is the 10th domain of human tepsin (5WF9).

We found that *Tg*TEP shows a steady-state localization at the trans-Golgi. Live imaging demonstrates that *Tg*TEP vesicles are highly mobile within the cell, reaching the periphery and even the residual body of the parasite. Since the protein's sole predicted domain suggests a possible involvement in vesicular trafficking, an inquiry into its exact location with regards to the vesicle cell membrane was done via the use of a construct expressing a cytoplasmic GFP-nanobody fused to a Halo-tag. The GFP-nanobody targeted to the cytoplasm was seen to colocalize exactly with the SYFP2-tagged protein of interest, thus confirming that the C-terminal SYFP2-tag also localized to the cytoplasm. Taken together, these results suggest that *Tg*TEP might not be directly associated with a specific Golgi or postGolgi compartment but instead requires the interaction with an adaptor complex (De Camilli et al., 2002). Tepsins have been identified as an accessory protein of the AP4 (Borner et al., 2012). Adaptor protein complexes selectively incorporate cargo proteins into nascent vesicles and recruit the transport machinery (Davies et al., 2018). While the biological role of APs has been well described in opisthokonts, their functions in apicomplexans are still largely unknown. For example, AP2 has been recently identified as a component of the micropore and food vacuole, structures involved in endocytosis in *T. gondii* and *P. falciparum*, respectively (Koreny et al., 2023), and AP1 appears to be required for vesicle transport to the rhoptries of the parasite (Kaderi Kibria et al., 2015; Klinger et al., 2024; Venugopal et al., 2017). Indeed, several components of the trafficking machinery,

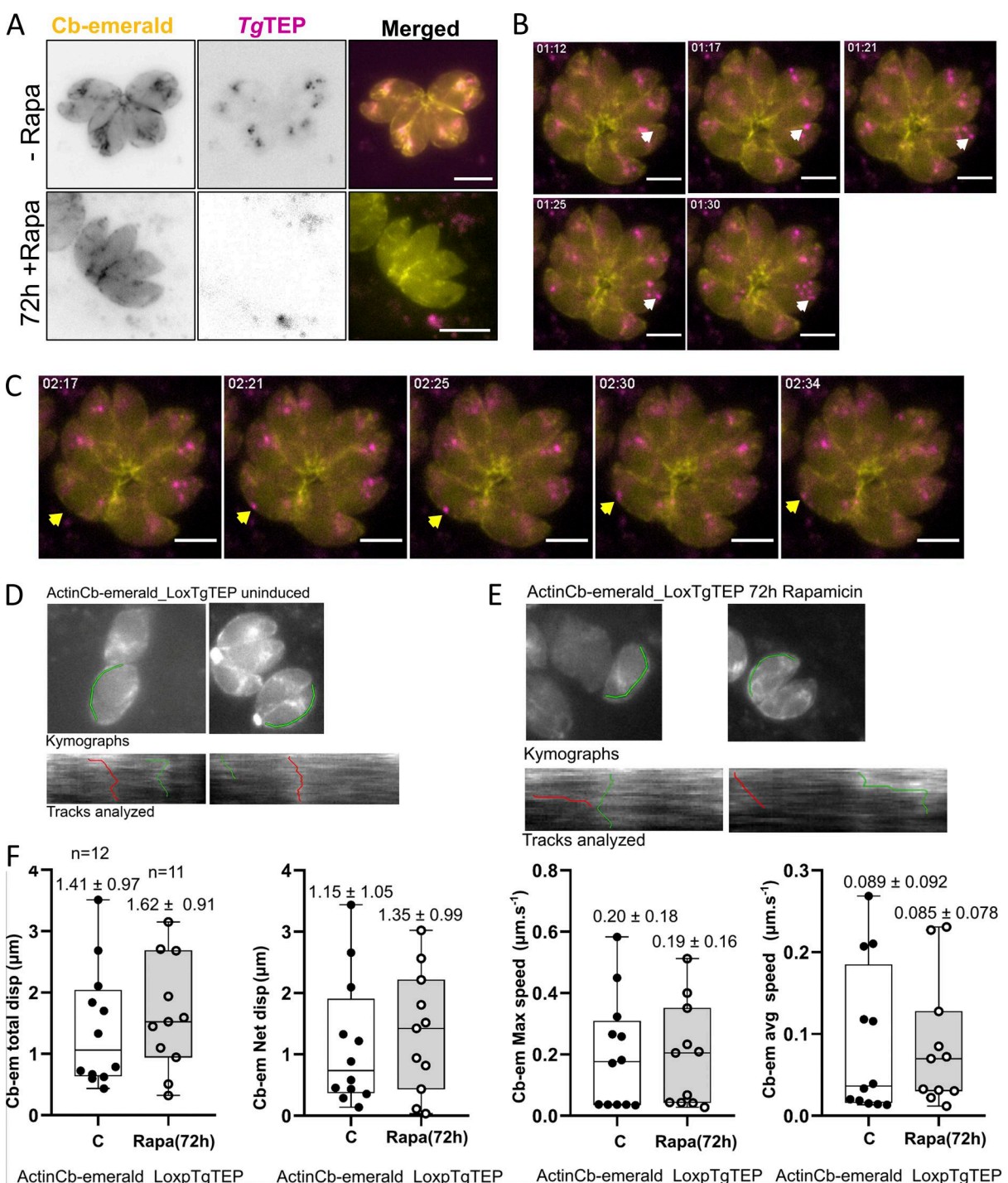

Figure 7. **TgTEP interacts with the actomyosin system. (A)** Immunofluorescence images suggested that knockout of *Tg*TEP (in magenta) resulted in a change in actin filament formation (chromobody-emerald in yellow). In WT parasites, the filaments primarily localize at the actin polymerization center near the Golgi body and connect the parasites within the parasitophorous vacuole at the basal end. Upon knockout of *Tg*TEP, less actin filaments were observed at the actin nucleation center around the Golgi, and the filaments connecting the parasites appeared more prominent. **(B and C)** In live movies (see Video 1), *Tg*TEP (in magenta) was seen colocalizing and moving along actin filaments (marked with the Cb-emerald in yellow). **(B)** Still images taken from the live movies wherein the white arrow indicates vesicles that are moving along actin filaments close to the actin nucleation center. **(C)** The yellow arrow points toward vesicles, which are moving along actin filaments along the periphery of the parasites. **(D and E)** Representative kymographs and analyzed tracks of inducible *Tg*TEP KO in actin-chromobody emerald-LoxP-TgTEP cell lines. Top panels show the ROI path (green) for kymograph analysis. Bottom panels show kymograph with tracks (green and red) chosen for extracting quantitative data. **(D and E)** The left panel shows an untreated parasitophorous vacuole, while (E) shows an example of a vacuole 72 h after induction with rapamycin. **(F)** Actin kinetics estimated as a measurement of actin chromobody displacement support no changes in actin kinetics as a result of abrogation of *Tg*TEP. KO, knockout.

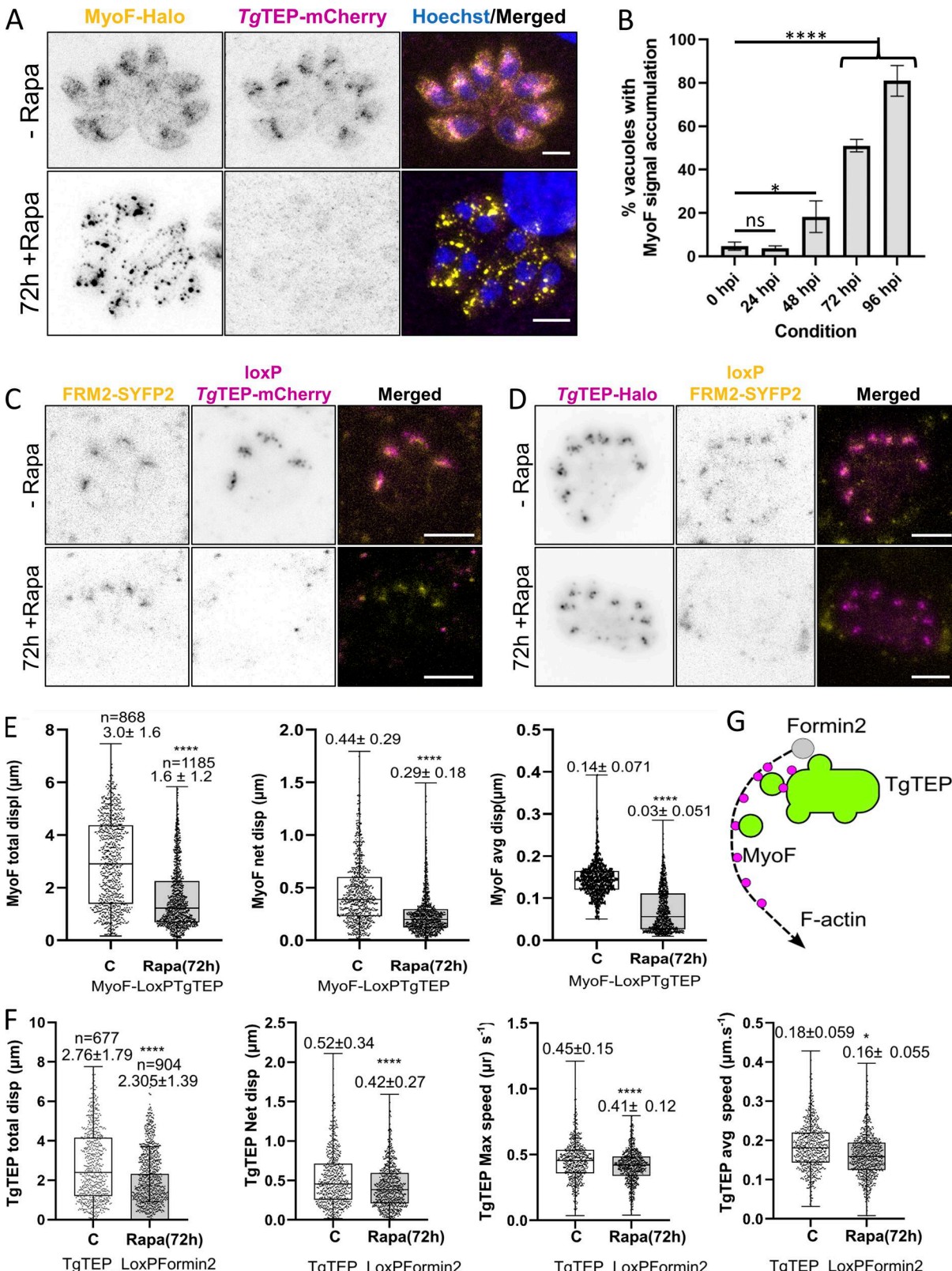

Figure 8. **_Tg_TEP knockout alters MyoF dynamics but has no effect on FRM2. (A)** In WT parasites, MyoF is seen to localize around the periphery of the cells and near the actin nucleation center proximal to the Golgi body. Upon knockout of _Tg_TEP (in magenta; + 72 h Rapa), MyoF (in yellow) was seen to form aggregates within the parasites. This was observed in both live as well as fixed parasites. MyoF and _Tg_TEP images were recorded with STED, while the nuclei (labelled with Hoechst) were taken with the confocal setting. Scale bars are 3 µm. **(B)** The number of parasitophorous vacuoles with altered MyoF localization was seen to be significantly higher compared with WT parasites starting at 48 h post-induction (hpi) with 50 nM rapamycin. The assay was done thrice, with a minimum of 100 vacuoles quantified per condition per replicate. Data are plotted as mean ± SD. One-way ANOVA with Tukey's multiple comparison test was

applied, with P values being represented as follows: ns ≥ 0.05; * = 0.01–0.05; **** ≤0.0001. **(C)** FRM2, typical localizing at the Golgi region, seemed unaffected upon knockout of *Tg*TEP. **(D)** Knockout of FRM2 has no influence on the location of *Tg*TEP at the TGN. **(E)** KO of *Tg*TEP (72 hpi) hampered transport kinetics of MyoF. Estimated displacement (total and net displacement) of MyoF in the PV was significantly affected. **(F)** Inducible FRM2 KO cell line showed that *Tg*TEP-dependent traffic was affected 72 hpi with rapamycin, suggesting a role of actin regulating the distribution of *Tg*TEP. **(G)** Proposed model: *Tg*TEP vesicles depend on actin polymerization for directional trafficking, mediated by MyoF. Upon *Tg*TEP deletion, MyoF accumulates in immobile aggregates, whereas FRM2 positioning remains unchanged. All scale bars except in A are 5 μm. KO, knockout.

including the epsilon subunit of AP4, have been identified in proximity labelling and co-IP.

Conditional disruption of *Tg*TEP resulted in parasite death, and, in line with its localization at the trans-Golgi, parasites depleted for *Tg*TEP showed a severely affected Golgi body structure resulting in its swelling and vesiculation. Interestingly, this effect was limited to the trans-Golgi and the Golgi compartments marked by DrpB and syntaxin-6, while no effect was seen on the cis-Golgi or the ELC marked by GRASP-RFP and ProM2AP, respectively. Surprisingly, despite this highly vesiculated trans-Golgi, this did not appear to affect the trafficking of secretory proteins such as the micronemes, the rhoptries, and the dense granules. Other structures such as the IMC and the apicoplast were similarly observed to be largely unaffected following knockout of *Tg*TEP. In contrast to this, using CPL as a marker for the PLVAC revealed a defect in its biogenesis. Upon knockout of *Tg*TEP, CPL, which typically localizes to vesicles diffused throughout the cytoplasm (Miranda et al., 2010; Parussini et al., 2010), was seen to accumulate at the Golgi, indicating that cargo arrest of PLVAC material in the trans-Golgi is the primary phenotype. This accumulation was seen to peak around 48 h after induction of the knockout, prior to the fragmentation of the Golgi body, which starts to occur at around 72 h. A recent publication (He et al., 2025) showed a similar defect in PLVAC formation in the absence of *Tg*TEP. While there are broad similarities in our findings, we note key differences between our findings and those of He et al. (2025). While they propose that *Tg*TEP localizes to the PLVAC, our data, along with findings from the Bradley Lab (Pasquarelli et al., 2024), strongly support a trans-Golgi localization. Furthermore, they report defects in invasion and gliding following *Tg*TEP depletion, phenotypes we do not observe in our system. These discrepancies may arise from the different experimental strategies employed.

We therefore conclude that *Tg*TEP is required for a highly defined trafficking pathway leading from the TGN to the PLVAC. Disruption of this pathway leads to vesiculation of the trans-Golgi, though other trafficking pathways are still functional. Given the diversity of functions carried out by the PLVAC, including digestion of cellular proteins and endocytosed material (McGovern et al., 2018) and autophagy (Di Cristina et al., 2017), other downstream phenotypes observed, such as the collapsing of the mitochondrion, could be explained. Previous reports have shown similar effects of mitochondria morphology after starvation (Ghosh et al., 2012) or inhibition of autophagy factors (Varberg et al., 2018).

Using co-immunoprecipitation and proximity labelling via TurboID (Branon et al., 2018; Cho et al., 2020), we identified *Tg*TEP as an accessory protein of AP-4 and demonstrated that localization of AP-4 to the trans-Golgi depends on the expression of *Tg*TEP. In addition, proximity labelling revealed a link between *Tg*TEP, the cytosolic actomyosin system, and potentially the micropore, since components of the micropore (K13, Eps15L, and CGAR) have been identified, suggesting a link between AP4 trafficking and endocytosis. This is in good agreement with a recent study, implicating MyoF function with a role in endocytosis in *P. falciparum* (Schmidt et al., 2023). Furthermore, previous data have shown Golgi and post-Golgi vesicles to be dependent on MyoF for their trafficking (Carmeille et al., 2021), potentially linking *Tg*TEP/AP4 with MyoF-dependent trafficking that intersects with endocytosis. However, our study reveals that *Tg*TEP is not directly involved in endocytosis but might be essential for the digestion of uptake material. In line with this observation, knockout of *Tg*TEP was observed to result in abnormal aggregation of MyoF, while it did not seem to have any influence on FRM2 localization. Taken together, the data suggest that *Tg*TEP/AP4-specific transport depends on MyoF and F-actin, but not vice versa (see Fig. 10 for model).

In summary, based on our data, *Tg*TEP is hypothesized to be involved in the specific vesicle transport from the TGN to the PLVAC, potentially fusing with vesicles derived from the micropore, the site of parasite endocytosis (Koreny et al., 2023).

## Materials and methods
### Molecular evolution and structural modelling
#### Genomic and transcriptomic data collection
Transcriptome and genomic datasets were collected from publicly available sources at the National Centre for Biotechnology Information, and protein datasets were collected from EukProt V3 (Richter et al., 2022). 53 species were chosen, representing the best curated protein sequence datasets from all the known taxonomic classes of pan-eukaryotic species (TCS dataset). This was supplemented by 10 apicomplexan species to focus on this lineage. BUSCO scores for the quality of the protein datasets are provided in Table S1.

#### Homology searches and determination of conserved domains
Sequence searches for adaptor complexes (AP1–4), epsin, and tepsin were conducted using Analysis of Molecular Evolution and Batch Entry (AMOEBAE) (Barlow et al., 2023). Initial homology searches were conducted using single query protein sequences from *Homo sapiens* for pan-eukaryotic analysis and *P. falciparum* for apicomplexan searches. Forward searches were conducted using BLASTp and tBLASTn with an e-value cutoff of 0.05; the AMOEBAE pipeline then used all candidates meeting this threshold as queries for a search into the relevant query database (i.e., a reverse search) to confirm orthology. Based on

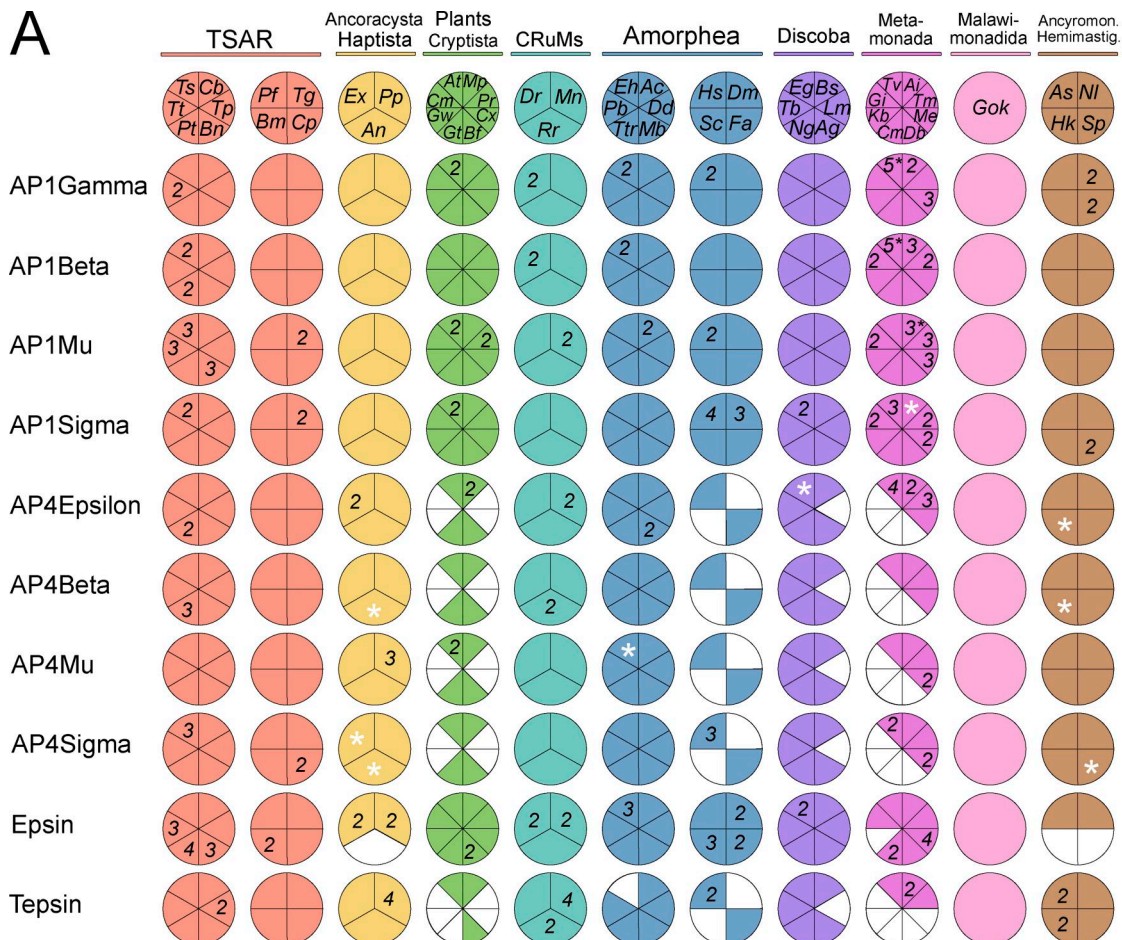

*Key*: **TSAR** - *Ts; Telonemia subtilis, Cb; Cafeteria burkhardae, Tp; Thalassiosira pseudonana, Tt; Tetrahymena thermophila, Pt; Paramecium tetraurelia, Bn; Bigelowiella natans, Pf; Plasmodium falciparum, Tg; Toxoplasma gondii, Cp; Cryptosporidium parvum, Bm; Brevimastigomonas motovehiculus, An; Ancoracysta twista.* **Haptista** - *Ex; Emiliania huxleyi, Pp; Prymnesium parvum.* **Plants** - *At; Arabidopsis thaliana, Mp; Marchantia polymorpha, Pr; Porphyra purpurea, Cm; Cyanidioschyzon merolae, Gw; Gloeochaete wittrockiana, Cx; Cyanophora paradoxa.* **Cryptista** - *Gt; Guillardia theta, Bf; Baffinella frigidus.* **CRuMs** - *Dr; Diphylleia rotans, Mn; Mantamonas plastica, Rr; Rigifila ramosa.* **Amorphea** - *Eh; Entamoeba histolytica, Ac; Acanthamoeba castellanii, Dd; Dictyostelium discoideum, Mb; Mastigamoeba balamuthi, Ttr; Thecamonas trahens, Pb; Pygsuia biforma, Hs; Homo sapiens, Dm; Drosophila melanogaster, Fa; Fonticula alba, Sc; Saccharomyces cerevisiae.* **Discoba** - *Eg; Euglena gracilis, Bs; Bodo saltans, Lm; Leishmania major, Ng; Naegleria gruberi, Tb; Trypanosoma brucei, Ag; Andalucia godoyi.* **Metamonada** - *Tv; Trichomonas vaginalis, Ai; Anaeramoeba ignava, Tm; Trimastix marina, Me; Monocercomonoides exilis, Db; Dysnectes brevis, Cm; Carpediemonas membranifera, Kb; Kipferlia bialata, Gi; Giardia intestinalis.* **Malawimonadida** - *Gok; Gefionella okellyi.* **Ancyromonadida** - *As; Ancyromonas sigmoides, Nl; Nutomonas longa.* **Hemimastigophora** - *Hk; Hemimastix kukwesjijk, Sp; Spironema_sp.*
★ : Passed reverse blast but some paralogs were not identified. Numbers represent paralog count.
✿ : Failed reverse blast but identified in forward searches or reported in other publications.

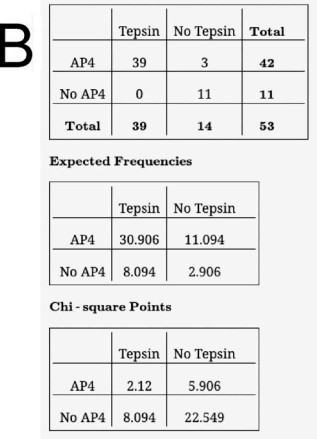

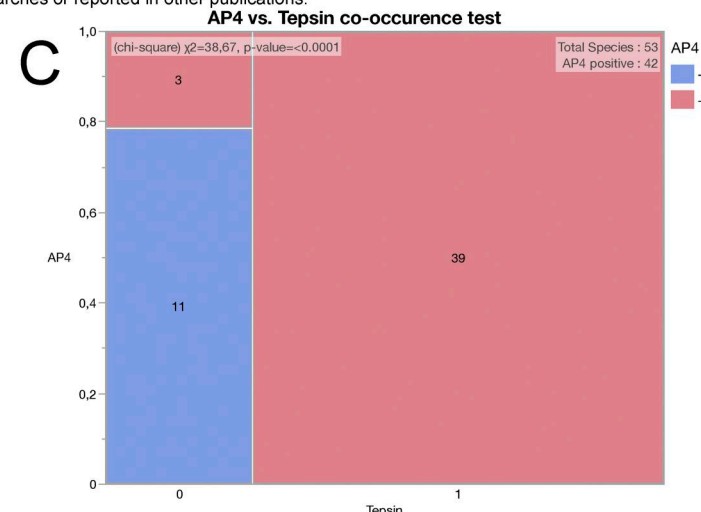

Figure 9. **Homology searches of adaptor complexes AP1 and AP4 components, epsin, and tepsin across 53 pan-eukaryotic species with statistical assessment for co-occurrence of AP4 and tepsin. (A)** Coulson plot representing result of HMMer search based comparative genomics conducted across

selected eukaryotic species with their eukaryotic classification. Circles or sections filled with color demonstrate presence of protein homologs; unfilled sections demonstrate loss. Color only represents single paralog identification; numbers represent multiple paralogs. Key is provided for the abbreviated species binomial names. **(B)** Independence chi-square test calculation matrices for presence of absence of variables: AP4 and tepsin. First matrix shows actual values of different variations of presence or absence of both the variables, second matrix shows the expected frequencies for the four variations, and third matrix represents the chi-square points, along with the calculated $\chi^2$ and P value. **(C)** Graphical representation of the AP4 and tepsin co-occurrence test is provided with the calculated $\chi^2$ and P value, along with the blocks of graph representing all the variations of occurrence.

the initial searches, hidden Markov models (HMMs) were built to perform HMMer searches in protein datasets of 53 pan-eukaryotic species and 10 apicomplexan species using AMOE-BAE; positive hits were then reverse searched into *Naegleria gruberi* and *P. falciparum*. Output of the HMMer searches are provided in Tables S1, S2, and S3. The confirmed homologs were subjected to conserved domain analyses using InterProScan (Paysan-Lafosse et al., 2022) and confirmed with phylogenetic analyses. Pan-eukaryotic co-occurrence or coevolution chi-square test for AP4 and tepsin was conducted manually; expected frequencies and chi-square points for the calculation of chi-square ($\chi^2$) value and P value are provided in Fig. 9 B. Graphical representation of the same and chi-square calculation was conducted using a statistics software "JMP 18" (https://www.jmp.com/en_us/software/new-release/new-in-jmp.html).

### Phylogenetic analysis

All the positive homologs for each protein were aligned using MAFFT v7.505. Alignments were visualized using AliView (Larsson, 2014). Partial sequences were removed from the final analyses. All the alignments are available upon request. Alignments were automatically trimmed to conserve the informative regions using Block Mapping and Gathering with Entropy (Criscuolo and Gribaldo, 2010). Maximum likelihood phylogenetic analysis was performed using IQ-TREE 2 (Minh et al., 2020), and the best fit models were chosen using IQ-TREE Model finder (Kalyaanamoorthy et al., 2017). Maximum likelihood and posterior probability support values were collected using nonparametric bootstrapping and Bayesian posterior probability calculation (-b -N 1,000 -alrt 1,000 -abayes) for phylogenetic analysis in Fig. 1 B. Ultrafast bootstrapping (-B -N 1000) support values were generated for other phylogenetic

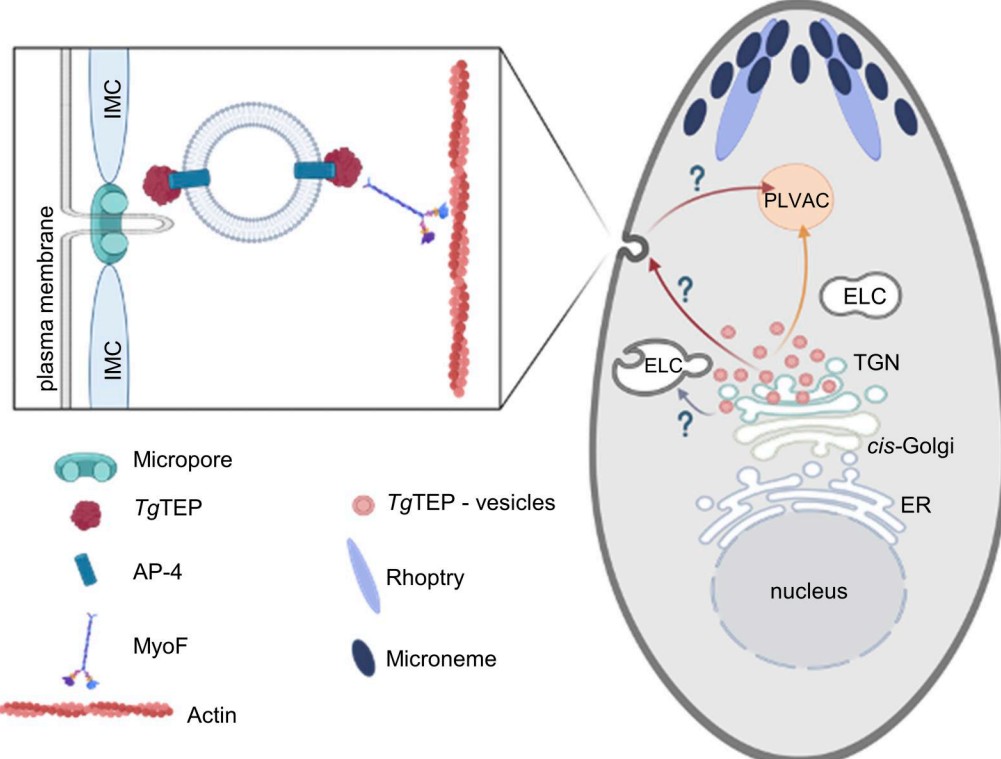

Figure 10. **Model for *Tg*TEP-dependent vesicular transport.** The schematic illustrates the role of *Tg*TEP in coordinating vesicle transport from the TGN to the PLVAC. *Tg*TEP, in complex with the AP4, binds to vesicles that bud from the TGN. These vesicles are transported along actin filaments in a MyoF-dependent manner (orange arrow). A subset of these *Tg*TEP/AP4-positive vesicles may fuse with endocytic vesicles originating from the micropore, contributing to the delivery of internalized material to the PLVAC (red arrow). Upon *Tg*TEP deletion, vesicle trafficking is disrupted, leading to the accumulation of vesicles at the TGN, fragmentation of the trans-Golgi, and impaired delivery of PLVAC cargo. As a result, the PLVAC fails to properly digest or recycle material, which may lead to secondary defects such as mitochondrial fragmentation. Other secretory pathways (e.g., to micronemes, rhoptries, or dense granules) remain functional and are shown as unaffected (gray arrows).

analyses (Data S1). Trees were visualized and assessed using FigTree v1.4.4.4. Accession numbers of all the sequences used for phylogenetic analyses are provided in the respective figures or Table S5.

### Structural analysis

Predicted 3D structures were obtained in Protein Data Bank (PDB) format from AlphaFold v2.0 (Jumper et al., 2021; Varadi et al., 2022) for AP4-ε of human (UniProt ID: Q9UPM8, pLDDT: 72.65) and *T. gondii* (UniProt ID: S8F423, pLDDT: 65) and tepsin of *T. gondii* (UniProt: S7VVX2, pLDDT: 45.88). Crystal structure of human tepsin ENTH (10th) domain PDB format file was obtained from the PDB server. 3D structures of predicted AP4-ε and *Tg*TEP, were super-imposed with predicted AP4-ε and crystal structure of 10th from human, respectively, in PyMOL 3.0 (https://pymol.org/). Quality of confidence for the structural alignments was confirmed using TM-align (Zhang and Skolnick, 2005). Human 10th crystal structure was used for the collection of secondary structure and alignment with the identified api-complexan protein sequences using ESPript 3.0 (Robert and Gouet, 2014).

### Culturing of *T. gondii* and host cells

*T. gondii* tachyzoites were cultured in human foreskin fibroblast (HFF; SCRC-1041; ATCC) monolayers at 37°C and 5% $CO_2$ in DMEM (D6546; Sigma-Aldrich) supplemented with 10% FBS (Bio&SELL FBS.US.0500), 4 mM L-Glutamate (G7513; Sigma-Aldrich), and 20 µg/ml gentamicin (G1397; Sigma-Aldrich).

### Generation of tagged and floxed parasite strains

New strains were generated via the use of CRISPR/Cas9 as previously described (Stortz et al., 2019). Guide RNAs targeting the regions of interest were designed using Eukaryotic Pathogen sgRNA Design Tool (EuPaGDT) (Peng and Tarleton, 2015). The sequences of all sgRNAs can be found in supplementary tables. The sgRNAs were synthesized as oligos by Thermo Fisher Scientific as described by Stortz et al. (2019) and annealed in buffer (10 mM Tris-base, pH 7.5–8, 50 mM NaCl, and 1 mM EDTA) at 95°C. Ligation of the annealed oligos into a vector coding for Cas9-YFP was done using T4 DNA ligase and incubated overnight. All sgRNA plasmids were confirmed by sequencing (Eurofins Genomics). For the modification of endogenous material, such as tagging and insertion of loxP sequences, the sgRNAs were designed to target the N or C terminus of the genes of interest. The repair templates containing the loxP sequences flanked by 33 bp of homology sequence were synthesized as single-stranded oligos from Thermo Fisher Scientific, whereas the ones encoding the tags were generated via PCR amplification from plasmids listed in supplementary tables; the 50 bp homology sequences flanking the tags being introduced into the template via the primers. For the insertion of exogenous material into the parasites, including the chromobody-emerald amplified from the plasmid used by Periz et al. (2017), GRASP-RFP was kindly gifted by Kristin M. Hager (Hager lab, University of Notre Dame, IN, USA) (Pfluger et al., 2005), and a GFP-nanobody-Halo construct amplified from a plasmid generated during this study, the sgRNA was designed to target the first exon of the UPRT locus (Li et al., 2022). The primers used to amplify these constructs were designed to have 50 bp of homology upstream of the UPRT cut site and 50 bp of homology downstream of the UPRT gene, thus replacing the gene but retaining both UTRs.

All PCR products amplified were purified using a PCR purification kit from Blirt (Blirt; EM26.1), and all templates were pooled with 10–12 µg of plasmids carrying the corresponding sgRNAs. These pools were then ethanol precipitated and transfected into freshly egressed RHDiCredKu80 tachyzoites created by Hunt et al. (2019) as by Stortz et al. (2019). The Amaxa 4D-Nucleofector system (AAF-1003X; Lonza) was used for all transfections, with both DNA and parasites mixed in P3 buffer and electroporated in 100-µl cuvettes (P3 Primary cells 4D-Nucleofector X kit L, V4XP-3024; Lonza) using the program FI-158. The transfected parasites were allowed to invade fresh HFFs and replicate for 48 h, after which they were manually egressed and filtered through a 3-µm filter. FACS (FACSARIA III; BD Biosciences) was used to enrich for those expressing Cas9-YFP, these being transferred to 96-well plates, thus generating clonal populations. Proper integration of repair templates was confirmed via PCR and sequencing (Eurofins Genomics).

### Plaque assays

$5 \times 10^2$ parasites per well were used to infect confluent HFFs in a 6-well plate. 50 nM of rapamycin per well was used for induction of the knockouts, while an equal volume of DMSO was used as the non-induced vehicle control. The plates were left to incubate undisturbed for 7 days, after which the cells were fixed with 100% ice-cold methanol for 20 min at room temperature and washed with PBS. The cells were then stained with Hemacolor Rapid Staining for Blood Smear solution 2 for 1 min and solution 3 for 2 min and finally washed thoroughly with water. Images of the plaque assays were taken using a 10× objective on a Leica Dmi8 widefield microscope attached to a Leica DFC9000 GTC camera and the associated LasX Navigator software. A 12 × 12 field area at the center of the wells was chosen. Focus maps and autofocus settings were employed. Following the acquisition of the images, the "mosaic merge" function on the LasX software was used to generate a merged image from the multiple fields of view imaged. Three biological replicates were done.

### Invasion, replication, and egress assays

For invasion assays, $3 \times 10^6$ parasites were incubated for 48 h in the presence of 50 nM rapamycin, the length of time necessary for the protein of interest to no longer be visible via IFA. These were then egressed manually, after which $5 \times 10^5$ parasites were used to infect fully confluent HFFs on coverslips. The cells inoculated with parasites were incubated on ice for 10 min to allow the parasites to settle on the cells, after which these were then incubated at 37°C for 30 min to allow the parasites to invade. The coverslips were then fixed with 4% PFA, after which the parasites were stained to allow visualization and distinction between the invaded and extracellular parasites. Extracellular parasites were labelled by incubating the slides with antibodies against SAG1 diluted in PBS and 3% BSA for 1 h at room temperature. The cells were then washed and labelled with secondary antibodies for 1 h in the dark. Following the labelling of extracellular

parasites, the intracellular parasites were labelled by following the same protocol to label GAP45, also incorporating Triton X-100.

For replication assays, the parasites were prepared in the same way as for invasion assays. $5 \times 10^5$ parasites were used to infect fully confluent HFFs on coverslips, after which these were allowed to invade for 1 h at 37°C. The coverslips were then washed thoroughly thrice using fresh supplemented DMEM, after which the parasites were allowed to replicate for 24 h. Following this incubation, the cells were fixed using 4% PFA, and the parasites were labelled using anti-GAP45 antibodies.

For egress assays, parasites were incubated in the presence or absence of 50 nM rapamycin or DMSO for 24 h. Following incubation, the parasites were egressed manually and $1 \times 10^5$ used to infect confluent HFFs on coverslips. The parasites were allowed to invade for 4 h, after which the slides were washed thrice and fresh media added. After an incubation period of 36 h, egress was induced or not induced via the addition of 2 μM calcium ionophore (Ci A23187) in supplemented DMEM without FCS. Induction of egress was done for 5 min at 37°C, after which the slides were fixed in 4% PFA for 15 min. Staining for egressed parasites was done as was done for invasion assays.

A minimum of 100 vacuoles per replicate were counted, with three biological replicates done for all assays.

### Immunofluorescence assays

Parasites were incubated in fresh HFFs at 37°C and 5% $CO_2$ for the desired length of time, after which they were either imaged live or fixed with 4% PFA for 15 min at room temperature. Live imaging was generally done for parasites which expressed proteins tagged with self-labelling tags such as SNAP-tag and Halo-tag. For this labelling, synthetic dyes were added according to the manufacturer's instructions. In the case of samples expressing Halo- or SNAP-tagged proteins which required fixation, the synthetic dyes were added prior to fixation. In the case of fixed IFA slides, cells were washed thrice with PBS following fixation, permeabilized, and blocked for 45 mins at room temperature with 3% BSA and 0.2% Triton X-100 in PBS, then labelled for 1 h at room temperature with primary antibodies diluted in the blocking/permeabilizing buffer. The cells were then again washed thrice with PBS and labelled for 1 h at room temperature in the dark with secondary antibodies diluted in the blocking/permeabilizing buffer. All antibodies are listed in supplementary tables. DNA staining was done by incubating with 0.4 μM Hoechst for 5 min at room temperature in the dark. Mounting of slides was done with ProLongTM Gold antifade mountant (Thermo Fisher Scientific).

For IFAs using BFA, cells were incubated in the presence of 100 μg/ml of the drug for 1 h prior to fixation. For visualization of the localization of biotinylated proteins, 150 μM biotin was added for either 30 min or 6 h prior to fixation.

### SAG1 endocytosis assay

The SAG1 endocytosis assay was done as has been described by Koreny et al. (2023). Briefly, SAG1 was tagged with a Halo-tag upstream of the GPI-anchor. These parasites also expressing tagged and floxed *Tg*TEP were incubated in the presence or absence of 50 nM rapamycin for 24 h, after which these were mechanically released and then incubated in the presence of the membrane non-permeable dye Alexa-488 for 1 h. After removal of the excess dye with washes, the parasites were transferred unto host cells and allowed to replicate for a further 24 h before imaging and analysis. The percentage of vacuoles with vesicles labelled with SAG-1 halo bound to the membrane non-permeable dye was then calculated.

### Microscopy

Widefield microscopy was done using a Leica Dmi8 widefield microscope attached to a Leica DFC9000 GTC camera. Z-stack images were obtained using a 100× objective, while images for phenotype quantification purposes were taken using a 63× object.

All time-lapse video microscopy was similarly done using the widefield microscope. The parasites were placed in a preheated chamber at 37°C and 5% $CO_2$. Fully supplemented DMEM FluoroBrite was used.

Confocal and STED microscopy were performed using the Abberior 3D STED microscope, equipped with 3 color STED. Imaging settings were adjusted according to the signal strength of every sample to optimize signal to noise and resolution. Typically, 60 × 60 × 250 nm sampling was performed for confocal mode, while 30 × 30 × 250 nm was done for 2D STED.

### Electron microscopy

Parasites were incubated in the presence or absence of 50 nM rapamycin for 48 or 72 h, after which these were fixed using 2.5% glutaraldehyde in 0.1 M phosphate buffer, pH 7.4. Briefly, the cells were washed thrice with PBS, after which these were postfixed with 1% (wt/vol) osmium tetroxide for 1 h. After washing with PBS and water, the samples were then stained en bloc with 1% (wt/vol) uranyl acetate in 20% (vol/vol) acetone for 30 min and dehydrated in a series of graded acetone and embedded in Epon 812 resin. Sections of 60 nm thickness were cut using a diamond knife on a Reichert Ultracut-E ultramicrotome. These sections were then mounted on collodium-coated copper grids, stained with 80 mM lead citrate (pH 13), and imaged using an EM 912 transmission electron microscope (Zeiss) equipped with an integrated OMEGA energy filter operated in the zero-loss mode at 80 kV. Images were taken using a 2k × 2k slow-scan CCD camera (Tröndle Restlichtverstärkersysteme).

### Image processing

Leica LasX software (v. 3.4.2.183668) and Imspector (v.16.3.19714-w2408; Abberior) were used for image acquisition at the Leica widefield microscope and at the Abberior 3D STED microscope, respectively. LI-COR Image Studio was used to acquire western blot images. All subsequent image processing and analysis were done using Fiji (ImageJ) software v.2.1.0 (Schindelin et al., 2012) and/or Icy Image Processing Software 1.8.6.0. Images taken for the purposes of phenotype quantification were not processed.

Relative fluorescence or corrected total cell fluorescence (CTCF) was calculated as follows:

CTCF = Integrated density - (area of selection × mean fluorescence of background).

### Analysis of actin, *MyoF*, *formin2*, and *Tg*TEP dynamics

To investigate the role of *Tg*TEP ko in MyoF dynamics, cell line MyoF-LoxPTgTEP was analyzed before (control) and 72 h after ko induction with rapamycin. At least 20 movies (34 frames, 1.3-s time interval) were taken for each condition. For the control experiment, 20 PV were further analyzed (stages between 1 and 2): 16 PV (stage 1); 4 PV (stage 2). For MyoF-LoxPTgTEP (treated with rapamycin for 72 h), 23 vacuoles analyzed (stages between 1 and 32) 1 PV (stage 1); 9 PV (stage 2); 7 PV (stage 4); 1 PV (stage 32).

To assess the effect of formin knockout on *Tg*TEP dynamics, *Tg*TEP trafficking was analyzed in the *Tg*TEP_LoxPFormin2 cell line before (control) and 72 h after induction with rapamycin. For each condition, 20 time-lapse movies were recorded, consisting of 34 frames with a time interval of 0.8 s between frames.

In the control condition, a total of 40 vacuoles were analyzed across developmental stages 1–8, with 20 vacuoles at stage 1, 16 vacuoles at stage 2, three vacuoles at stage 4, and one vacuole at stage 8. Following rapamycin induction for 72 h, 30 vacuoles were analyzed in the TgTEP_LoxPFormin2 cell line, with seven vacuoles at stage 1, 15 vacuoles at stage 2, five vacuoles at stage 4, and three vacuoles at stage 8.

To examine actin chromobody dynamics, the inducible knockout cell line ActinCb-emerald_LoxP_TgTEP was used. Measurements were performed under two conditions: without rapamycin treatment (control) and after 72 h of rapamycin induction. For each condition, 10 time-lapse movies were recorded. Six vacuoles were selected for detailed analysis in each condition, with movies consisting of 34 frames captured at 1.6-s intervals, focusing on vacuoles in stages 1 and 2. Trafficking movies were analyzed using the Icy platform for bioimage analysis (de Chaumont et al., 2012). PV for analysis particle displacement was selected based on good signal to noise, no drift with the vacuoles remaining in focus during the duration of the movie. Region of interest (ROI) tracks were drawn in places of particle movement and then analyzed by the KymographTracker plugin in Icy (Periz et al., 2019).

For the analysis, five pixels were selected for the width of the analyzed tracks. The kymogram was made without separating particle direction, and the analysis proceeded on at least two ROI tracks that defined the particle motion. The information was quantified by the Icy TrackManager plugin "MotionProfiler." The extracted values were then exported to Microsoft Excel for further analysis. The manual for the use of the plugin and kymograph analysis can be found on Icy website: http://icy.bioimageanalysis.org/plugin/KymographTracker.

### TurboID sample preparation

Biotin was added to the cells at a concentration of 150 µM for either 30 min or 6 h, after which the parasites were egressed manually and filtered using 3-µm filters on ice. The egressed parasites were washed with cold PBS on ice, pelleted at 1,500 × *g* for 10 min, and stored at –80°C until needed. $3 \times 10^7$ parasites and $2 \times 10^7$ parasites were used for the 30-min and 6-h time points, respectively. Dynabeads MyOne Streptavidin T1 bead preparation was done as recommended by the manufacturer (Invitrogen). 50 and 85 µl of beads were used for the 30-min and

6-h time points, respectively. Parasite pellets were lysed with 100 µl RIPA buffer (50 mM Tris-HCl [pH 8], 0.5% sodium deoxycholate, 150 mM NaCl, 1 mM EDTA, 0.1% SDS, and 1% Triton X-100) containing 1% SDS and Pierce protease inhibitor at a concentration of 1:10. Lysis was done on ice for 45 min, after which the mixture was diluted with more RIPA buffer to bring the SDS concentration down to 0.1%. The lysis supernatant was used to resuspend the prepared Dynabeads, and the lysis–bead mixture was incubated at room temperature for 30 min while rotating. Following incubation, the beads were washed with 1 ml RIPA buffer without Triton X-100 for five times, using the DynaMag magnet to separate the beads from the supernatant. The beads were finally washed with 1 ml 50 mM (pH 8) Tris-HCL three times and resuspended in 200 µl of Tris-HCL. 10% of the resuspended beads were kept for western blot controls.

### co-IP assay

co-IP assays were performed to investigate the interaction between *Tg*TEP and the AP4 complex. $10^9$ parasites per condition were collected and filtered using 3-µm filters on ice. Parasites were washed three times in ice cold PBS and pelleted at 1,500 × *g* for 10 min. Parasites were resuspended in lysis buffer (150 mM Hepes, pH 7.4, 300 mM NaCl, 1% NP-40, 1 mM MgCl$_2$, 25 U/ml benzonase, and EDTA-free protease inhibitors) and incubated at 4°C in rotation for 1 h. 1% of the lysed parasites were kept as an input sample. ~10 µg of antibodies against the tags for *Tg*TEP-YFP (α-GFP; Roche) and AP4ε-3xHA (α-HA; Roche) were added to the lysed parasites and incubated at 4°C in rotation for 1 h. One tube was left without antibodies as negative control for non-specific binding on beads. 100 µl of magnetic beads Protein G Dynabeads (Thermo Fisher Scientific) were added to each of the conditions and were incubated at 4°C in rotation overnight. Next morning, beads were pulled on a magnetic rack, and 1% of the sample was collected as output. Beads were washed with washing buffer (150 mM Hepes, pH 7.4, 300 mM NaCl, and 0.1% NP-40) 5 times. A sixth wash was done with washing buffer without NP-40, and 1% of the supernatant was collected for analysis. 10% of the beads were taken for western blot analysis, and the rest were stored at –80°C until samples were send for mass spectrometry. This assay was performed in biological triplicates.

### Mass spectrometry

The protocol done was as by Singer et al. (2023). Briefly, the beads were incubated with 10 ng/µl of trypsin in 1 M urea and 50 mM of NH$_4$HCO$_3$ for 30 min. These were then washed with 50 mM of NH$_4$HCO$_3$, and the resulting supernatant was digested overnight with 1 mM DTT. The digested peptides were alkylated and desalted, after which these were injected into an Ultimate 3000 RSLCnano system (Thermo Fisher Scientific). The peptides were separated via HPLC using a 15-cm analytical column (75 µm ID with ReproSil-Pur C18-AQ 2.4 µm from Dr. Maisch) with a gradient going from 4 to 40% acetonitrile in 0.1% formic acid for 50 min. The resulting effluent of the HPLC was then electrosprayed into a Q Exactive HF (Thermo Fisher Scientific), which was used in a data-dependent mode to automatically switch between full scan MS and MS/MS acquisition. Survey full scan

MS spectra (from m/z 375–1,600) with resolution R = 60,000 at m/z 400 (AGC target of $3 \times 10^6$) were acquired. The 10 most intense peptide ions having charge states between 2 and 5 were then isolated to a target value of $1 \times 10^5$ and fragmented at 27% normalized collision energy. The conditions used for mass spectrometry were as follows: 1.5 kV spray voltage, no sheath and auxiliary gas flow, 250°C heated capillary temperature, and 33,000 counts ion selection threshold.

MaxQuant 1.6.14.0 was employed for the identification of proteins, and quantification by iBAQ was done using the following parameters: UP000005641_T. gondii_20220321.fasta, MS tol: 10 ppm, MS/MS tol: 20 ppm Da, peptide false discovery rate (FDR): 0.1, protein FDR: 0.01 min, peptide length: 7, variable modifications: oxidation (M), fixed modifications: carbamido-methyl I, peptides for protein quantitation: razor and unique, Min. peptides: 1, and Min. ratio count: 2. The MaxQuant iBAQ Z-score normalized values of the identified proteins were plotted on a volcano plot using Perseus (Tyanova et al., 2016). Any missing values were replaced (width: 0.3 and downshift: 4), the FDR was set to 0.05, and the S0 value was set to 0.1. The t test was used.

### Software

In silico cloning was performed using ApE. A plasmid editor (by M. Wayne Davis. V.2.0.53c) software. Sequencing results were analyzed using BioEdit v.7.2. Generation of sgRNAs was done in EuPaGDT (Peng and Tarleton, 2015). Schemes were created using Adobe InDesign, Microsoft PowerPoint, or BioRender. *Tg*TEP domain schematic was done using DOG (Domain Graph, v. 2.0; [Ren et al., 2009]). Image analysis was done using Fiji (ImageJ) software v.2.1.0 (Schindelin et al., 2012), while analysis of mass spectrometry data was done using Perseus 1.6.15.0 (Max Planck Institute of Biochemistry). AlphaFold (Jumper et al., 2021; Varadi et al., 2022) was used to visualize the protein structures, while STRING v11.5 (Szklarczyk et al., 2011) was used to predict the phylogeny of the proteins of interest. EuPaGDT was used to design sgRNAs, and ToxoDB Toxoplasma Informatics Resources (NIAID) was used as a reference information database (Gajria et al., 2008).

### Statistics and reproducibility

All data were plotted by Excel (Microsoft 365) or GraphPad Prism 8.2.1. Statistical analyses were carried out using GraphPad Prism, with one-tail ANOVA, multiple comparison T test, or unpaired two-tailed Student's t test being done as required.

All images presented here are representative, and all IFAs were repeated at least three times with same results.

### Online supplemental material

Fig. S1 shows the ESPript 3.0 output of multiple sequence alignment with secondary structure of the crystal structure of AP4-interacting ENTH domain. Fig. S2 shows that knockout of *Tg*TEP does not affect the localization of micronemes, rhoptries, dense granules, IMC, or apicoplast but causes mitochondrial fragmentation. Fig. S3 shows the biotinylated proteins identified by mass spectrometry. Table S1 shows the BUSCO scores of selected taxa in complete analysis. Table S2 shows adaptins (AP1–4

large, beta, medium, and small subunits) across pan-eukaryotic species—confirmed hits from HMMer searches. Table S3 shows epsin and tepsin HMMer searches in Pan-eukaryotic 196 species TCS (EukProt) dataset—all hits. Table S4 shows AP1, AP4, epsin, and tepsin HMMer searches—all hits in apicomplexan species. Table S5 shows abbreviated adaptin sequence accession numbers in phylogenetic analysis as backbone (Hirst et al., 2014). Table S6 shows TurboID differential abundance in parasites treated for 6 h versus 30 min with biotin. Table S7 shows TurboID differential abundance in parasites treated for 6 h versus 30 min with biotin (phenotypic score less than –1.5 and predicted location). Table S8 shows TurboID at 30 min with biotin when compared with parasites without biotin. Table S9 shows TurboID differential abundance in parasites treated for 30 min with biotin and non-treated parasites (phenotypic score less than –1.5 and predicted location). Table S10 shows TurboID at 6 h with biotin when compared with parasites without biotin. Table S11 shows TurboID differential abundance in parasites treated for 6 h with biotin and non-treated parasites (phenotypic score less than –1.5 and predicted location). Data S1 shows maximum likelihood phylogenetic analyses trees constructed using IQ-TREE 2 for apicomplexan adaptor complexes AP1, AP4 subunits (A–D), pan-eukaryotic tepsin (E), and pan-eukaryotic AP1-4 subunits (F–I). Data S2 shows quantification data.

### Data availability

The mass spectrometry proteomics data have been deposited to the ProteomeXchange Consortium via the PRIDE (Perez-Riverol et al., 2022) partner repository with the dataset identifier PXD057662.

## Acknowledgments

We thank all colleagues who contributed antibodies and reagents for this study. We thank the Heaslip Lab for helpful discussions regarding the role of MyoF. We thank Sonja Härtle and Marina Kohn for access to their FACS as well as the expert support.

This project was funded within the DFG Priority Programme SPP2225, Projects ME 2675/7-1 and JI 463/2-2. The STED microscope was funded via a DFG-equipment grant (INST 86/1831-1).

Author contributions: Janessa Grech: data curation, formal analysis, investigation, methodology, visualization, and writing—original draft, review, and editing. Abhishek Prakash Shinde: conceptualization, formal analysis, investigation, methodology, and writing—original draft, review, and editing. Javier Periz: conceptualization, data curation, formal analysis, investigation, methodology, validation, visualization, and writing—review and editing. Mirko Singer: conceptualization, methodology, and supervision. Simon Gras: formal analysis, investigation, methodology, validation, and writing—review and editing. Ignasi Forne: data curation, formal analysis, resources, and writing—review and editing. Andreas Klingl: investigation. Joel B. Dacks: conceptualization, project administration, resources, supervision, and writing—review and editing. Elena Jimenez-Ruiz: conceptualization, data curation, formal analysis, funding acquisition, investigation, methodology, project administration, resources,

supervision, validation, visualization, and writing—original draft, review, and editing. Markus Meissner: conceptualization, data curation, formal analysis, funding acquisition, investigation, methodology, project administration, resources, supervision, validation, visualization, and writing—original draft, review, and editing.

Disclosures: The authors declare no competing interests exist.

Submitted: 30 December 2023

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

# Supplemental material

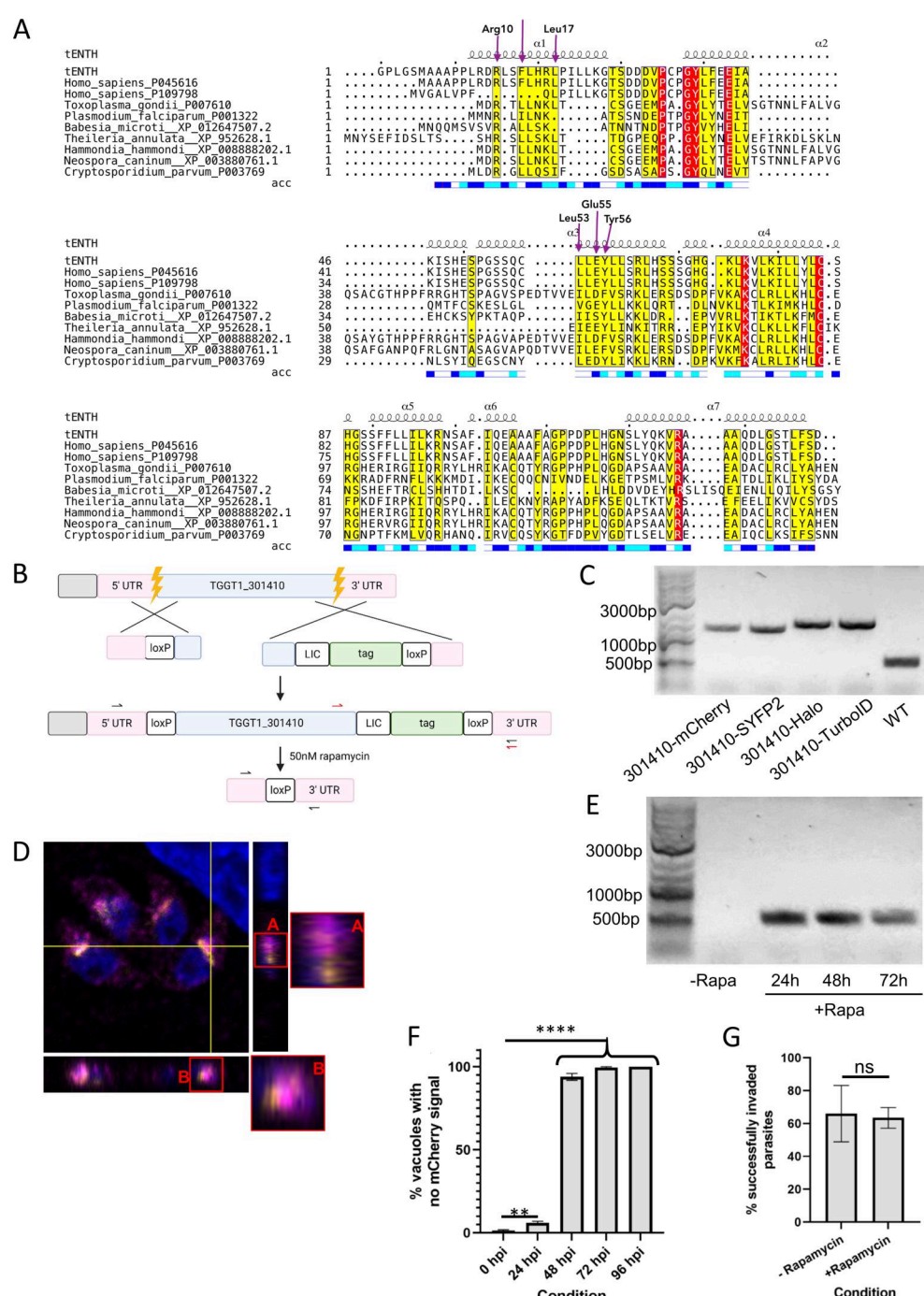

Figure S1. **ESPript 3.0 output of multiple sequence alignment (MSA) is shown with secondary structure of the crystal structure of AP4-interacting ENTH domain. (A)** Secondary structure element shown as squiggly lines represents the seven alpha helices of the crystal structure. Below the MSA is the bar representing solvent accessibility; blue, white, and teal colors represent accessible, buried, and intermediate residues of the protein. Residues highlighted in yellow demonstrate the alignment residue similarity; red highlight demonstrates identity. Purple arrows represent functional residues responsible for tepsin interaction with AP4. tENTH: *Toxoplasma* ENTH domain. **(B)** Tagging and floxing strategy for *Tg*TEP (TGGT1_301410). **(C)** Genotyping of the WT (524 bp) parasite strain as well as parasite strains obtained wherein TGGT1_301410 is endogenously tagged with mCherry (1290 bp), SYFP2 (1299 bp), Halo (1529 bp), and TurboID (1557 bp). Primer design corresponds to the red arrows in panel B. **(D)** Orthogonal views of the merged image in Fig. 2 C show that despite *Tg*TEP and SortLR come in very close proximity, they do not always colocalize. **(E)** Knockout of tagged TGGT1_301410 results in a band size of 514 bp. The floxed, endogenously tagged protein (-Rapa) could not be amplified due to its huge size. Primer design corresponds to black arrows in B. **(F)** The clone expressing *Tg*TEP-mCherry was used to quantify loss of protein via IFA. 95% of parasite vacuoles lost the protein by 48 hpi. Data are presented as mean ± SD. One-way ANOVA with Tukey's multiple comparison test was performed, with P values being represented as follows: ns ≥ 0.05; ** = 0.001–0.01; **** ≤0.0001. **(G)** Invasion assays were done, using parasites that were grown in the presence of 50 nM rapamycin or DMSO for a period of 48 h, after which these were manually released and allowed to invade fresh HFFs. All assays were done thrice, with a minimum of 100 parasites/vacuoles quantified per condition per replicate. All data are plotted as mean ± SD. Unpaired two-tailed Student's *t* test was where P = 0.8193. hpi, h post-induction. Source data are available for this figure: SourceData FS1.

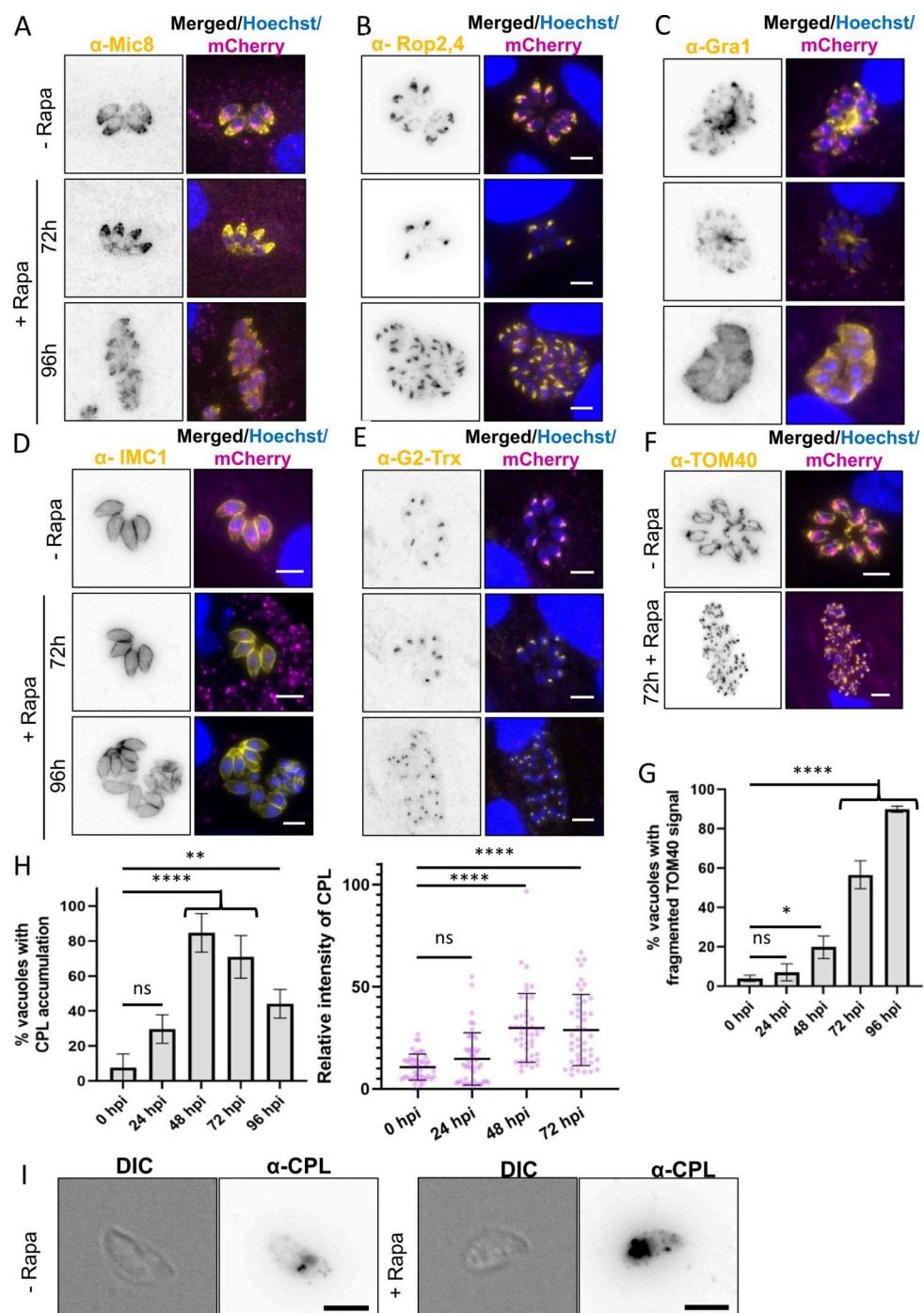

Figure S2. **Knockout of *Tg*TEP does not affect the localization of micronemes, rhoptries, dense granules, IMC, or apicoplast but causes mitochondrial fragmentation. (A–E)** IFAs using antibodies against (A) micronemes (Mic8), (B) rhoptries (Rop2,4), (C) dense granules (Gra1), (D) IMC (IMC1), and (E) apicoplast (G2-Trx) (all in yellow) showed that knockout of *Tg*TEP (tagged with mCherry, shown in magenta) has no effect on their localization. Immunofluorescence assays were done in triplicate, with a minimum number of 100 vacuoles per replicate observed. The nuclei were labelled with Hoechst. **(F)** Knockout of *Tg*TEP (in magenta) results in the fragmentation of the mitochondria (marked using the anti-TOM40 antibodies in yellow). All scale bars are 5 µm. **(G)** Quantification of mitochondrial fragmentation at different time points after induction of *Tg*TEP knockout. Data are presented as mean ± SD. One-way ANOVA with Tukey's multiple comparison test was done, with P values being represented as follows: ns ≥ 0.05; * = 0.01–0.05; **** ≤0.0001. **(H)** The percentage of parasitophorous vacuoles showing altered CPL localization was significantly higher in knockout mutants after 48 h post-induction (hpi) with 50 nM rapamycin. The assay was done three times, with a minimum of 100 vacuoles quantified per condition per replicate. The data are plotted as mean ± SD. Besides, CPL signal intensity quantifications confirmed an accumulation of CPL, this being significantly higher in knockout parasites after 48 h post-induction (hpi) with rapamycin. The assay was done three times, with the intensities of a minimum of 15 vacuoles quantified per condition per replicate. The data are plotted as mean ± SD, with the dots representing the value for each data point. For both assays, one-way ANOVA with Tukey's multiple comparison test was done. The P values are represented as follows: ns ≥ 0.05; ** = 0.001–0.01; **** ≤0.0001. **(I)** CPL signal, which typically appears more confined to a few structures in extracellular WT parasites, also appeared to accumulate in extracellular *Tg*TEP-knockout parasites. Scale bars are 3 µm.

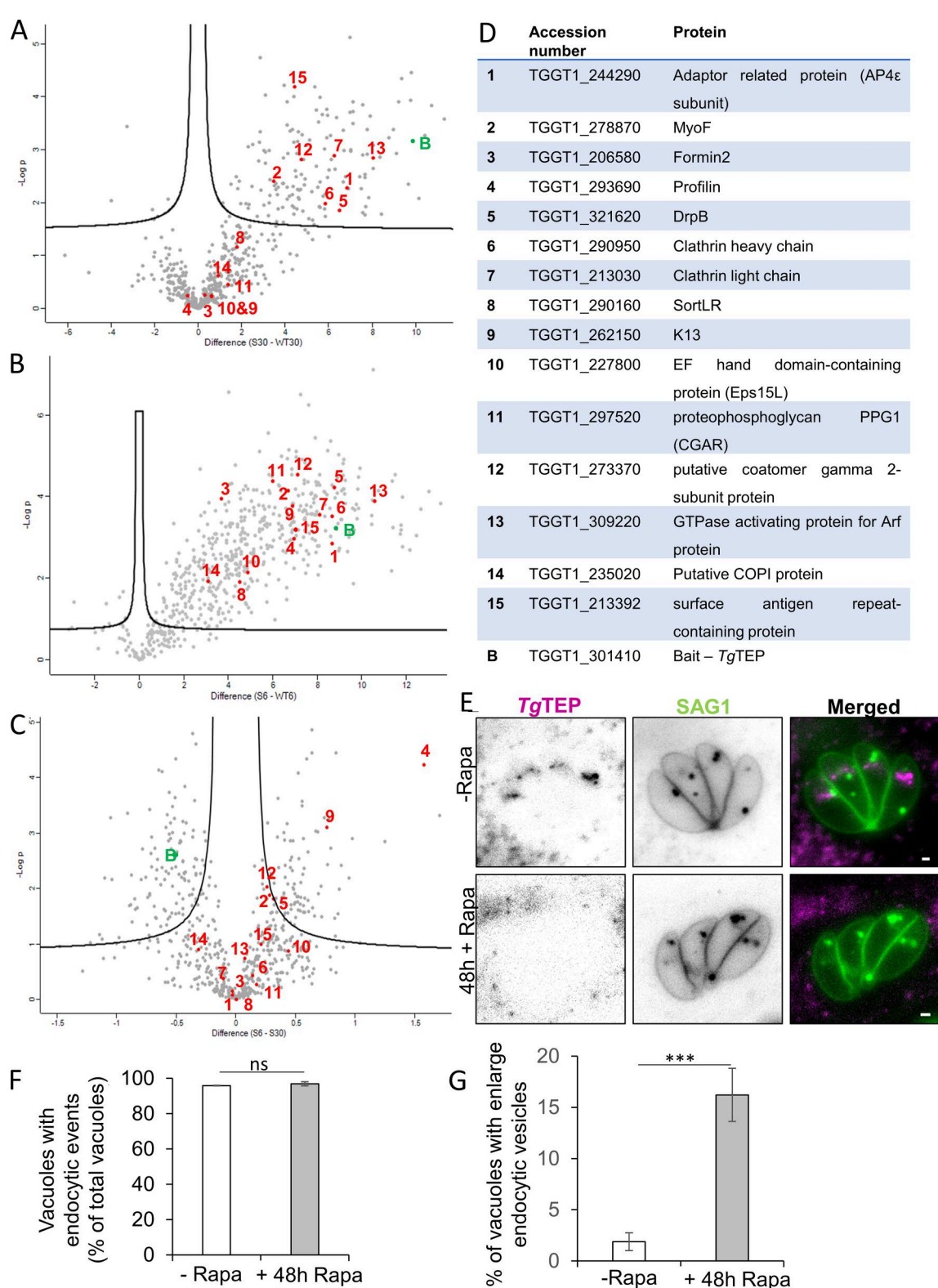

**D**

| | Accession number | Protein |
|---|---|---|
| 1 | TGGT1_244290 | Adaptor related protein (AP4ε subunit) |
| 2 | TGGT1_278870 | MyoF |
| 3 | TGGT1_206580 | Formin2 |
| 4 | TGGT1_293690 | Profilin |
| 5 | TGGT1_321620 | DrpB |
| 6 | TGGT1_290950 | Clathrin heavy chain |
| 7 | TGGT1_213030 | Clathrin light chain |
| 8 | TGGT1_290160 | SortLR |
| 9 | TGGT1_262150 | K13 |
| 10 | TGGT1_227800 | EF hand domain-containing protein (Eps15L) |
| 11 | TGGT1_297520 | proteophosphoglycan PPG1 (CGAR) |
| 12 | TGGT1_273370 | putative coatomer gamma 2-subunit protein |
| 13 | TGGT1_309220 | GTPase activating protein for Arf protein |
| 14 | TGGT1_235020 | Putative COPI protein |
| 15 | TGGT1_213392 | surface antigen repeat-containing protein |
| B | TGGT1_301410 | Bait – *Tg*TEP |

Figure S3. **Biotinylated proteins identified by mass spectrometry. (A and B)** show the difference in protein hits between WT (WT30) and *Tg*TEP-TurboID (S30) sample following 30 min of biotinylation, whereas (B) shows the difference between WT (WT6) and TurboID (S6) sample following 6 h of biotinylation. **(C)** shows the total number of proteins enriched during the 6-h experiment (S6) vs the 30-min experiment (S30) after normalization of protein abundance. Proteins of particular interest in panels A–C are numerated and listed in the table in D. **(E)** Halo-tagged SAG1 was labelled with a cell non-permeable dye for an hour, then parasites were allowed to replicate for a further 24 h to observe endocytosis of SAG1-containing vesicles. After addition of rapamycin, endocytosis was not affected. **(F)** Quantification of endocytosis events is similar in non-induced parasites (-Rapa) or induced parasites (+48 Rapa). **(G)** Quantification of vesicles demonstrates accumulation and enlargement upon deletion of *Tg*TEP. Scale bars are 5 µm. Data are plotted as mean ± SD. Unpaired two-tailed Student's *t* test was calculated for F and G, where P < 0.0001.

Video 1.  **TgTEP dynamics in intracellular *T. gondii* parasites.** Time-lapse widefield fluorescence microscopy of intracellular *T. gondii* parasites expressing TgTEP-mCherry (magenta) and the actin chromobody CbEm (yellow), which binds filamentous actin (F-actin). Parasites were not treated and are shown 24 h after infection in HFFs. TgTEP accumulates at the Golgi region and traffics dynamically within the cytoplasm along actin filaments. Imaging was performed using a Leica DMi8 widefield microscope with a 100× objective. Frames were acquired every 2.00 s and are displayed at 7 frames per second. Related still images are shown in Fig. 7, B and C.

Video 2.  **MyoF dynamics in LoxP-TgTEP parasites ± rapamycin.** Time-lapse widefield fluorescence microscopy of *T. gondii* LoxP-TgTEP parasites expressing endogenously tagged MyoF (gray). Parasites were treated with 50 nM rapamycin for 72 h (induced) or left untreated (non-induced) before imaging. The movie compares the dynamic localization of MyoF between the two conditions. In untreated parasites, MyoF shows directed movement; in contrast, MyoF becomes clustered and largely immobile following *Tg*TEP excision. Imaging was performed using a Leica DMi8 widefield microscope with a 100× objective. Frames were acquired every 1.39 s and are displayed at 7 frames per second. Related dynamics quantifications are shown in Fig. 8 E.

Video 3.  **Vesicle motility in LoxP-FRM2 parasites expressing TgTEP-Halo ± rapamycin.** Time-lapse widefield fluorescence microscopy of *T. gondii* LoxP-FRM2 parasites expressing endogenously tagged *Tg*TEP-Halo (gray). Parasites were treated with 50 nM rapamycin for 72 h (induced) or left untreated (non-induced). *Tg*TEP-positive vesicles show dynamic trafficking in the absence of FRM2 excision, whereas vesicle motility is reduced in FRM2-depleted parasites. Imaging was performed with a Leica DMi8 widefield microscope using a 100× objective. Frames were acquired every 1.39 s and are displayed at 7 frames per second. Related dynamics quantifications are shown in Fig. 8 F.

**Provided online is Table S1, Table S2, Table S3, Table S4, Table S5, Table S6, Table S7, Table S8, Table S9, Table S10, Table S11, Data S1, and Data S2. Table S1 shows the BUSCO scores of selected taxa in complete analysis. Table S2 shows adaptins (AP1–4 large, beta, medium, and small subunits) across pan-eukaryotic species—confirmed hits from HMMer searches. Table S3 shows epsin and tepsin HMMer searches in pan-eukaryotic 196 species TCS (EukProt) dataset—all hits. Table S4 shows AP1, AP4, epsin, and tepsin HMMer searches—all hits in apicomplexan species. Table S5 shows abbreviated adaptin sequence accession numbers in phylogenetic analysis as backbone (Hirst et. al. 2014). Table S6 shows TurboID differential abundance in parasites treated for 6 h versus 30 min with biotin. Table S7 shows TurboID differential abundance in parasites treated for 6 h versus 30 min with biotin (phenotypic score less than –1.5 and predicted location). Table S8 shows TurboID at 30 min with biotin when compared with parasites without biotin. Table S9 shows TurboID differential abundance in parasites treated for 30 min with biotin and non-treated parasites (phenotypic score less than –1.5 and predicted location). Table S10 shows TurboID at 6 h with biotin when compared with parasites without biotin. Table S11 shows TurboID differential abundance in parasites treated for 6 h with biotin and non-treated parasites (phenotypic score less than –1.5 and predicted location). Data S1 shows maximum likelihood phylogenetic analyses trees constructed using IQ-TREE 2 for apicomplexan adaptor complexes AP1, AP4 subunits (A–D), pan-eukaryotic tepsin (E), and pan-eukaryotic AP1-4 subunits (F–I). Data S2 shows quantification data.**

