## [Peer Review File · The Journal of Cell Biology]

Tepsin and AP4 Mediate Transport from the trans-Golgi to the Plant-Like Vacuole in Toxoplasma

Janessa Grech, Abhishek Shinde, Javier Periz, Mirko Singer, Simon Gras, I. Forné, Andreas Klingl, Joel Dacks, Elena Jimenez-Ruiz, and Markus Meissner

Corresponding Author(s): Markus Meissner, Ludwig-Maximilians-University Munich

Review Timeline:

Submission Date:	2023-12-30
Editorial Decision:	2024-01-31
Revision Received:	2025-03-13
Editorial Decision:	2025-04-21
Revision Received:	2025-06-06

Monitoring Editor: Elizabeth Miller

Scientific Editor: Andrea Marat

Transaction Report:

DOI: <https://doi.org/10.1083/jcb.202312109>

January 31, 2024

Re: JCB manuscript #202312109

Prof. Markus Meissner
Ludwig-Maximilians-University Munich
Veterinary Sciences, Experimental Parasitology
Lena-Christ-Straße 48
Munich 81249
Germany

Dear Prof. Meissner,

Thank you for submitting your manuscript entitled "Toxoplasma Tepsin is essential for actin-dependent transport from the trans-Golgi to the vacuole". The manuscript has been evaluated by expert reviewers, whose reports are appended below. Unfortunately, after an assessment of the reviewer feedback, our editorial decision is against publication in JCB.

You will see that the reviewers have substantial concerns that data has been interpreted in a subjective manner, along with issues with the writing and data presentation. We agree that given the limitations they have outlined, your current study does not conclusively show a role for TgTEG in trafficking as proposed. As JCB is interested in the conservation and function of these machineries in non-model organisms, if you are able to comprehensively address the reviewers' concerns, we would be willing to reconsider such a revised study.

However, I feel that the points raised by the reviewers are more substantial than can be addressed in a typical revision period. If you wish to expedite publication of the current data, it may be best to pursue publication at another journal.

Given interest in the topic, I would be open to resubmission to JCB of a significantly revised and extended manuscript that fully addresses the reviewers' concerns and is subject to further peer-review. If you would like to resubmit this work to JCB, please contact the journal office to discuss an appeal of this decision or you may submit an appeal directly through our manuscript submission system. Please note that priority and novelty would be reassessed at resubmission.

Regardless of how you choose to proceed, we hope that the comments below will prove constructive as your work progresses. We would be happy to discuss the reviewer comments further once you've had a chance to consider the points raised in this letter. You can contact the journal office with any questions at cellbio@rockefeller.edu.

Thank you for thinking of JCB as an appropriate place to publish your work.

Sincerely,

Elizabeth Miller, PhD
Monitoring Editor

Andrea L. Marat, PhD
Senior Scientific Editor

Journal of Cell Biology

Reviewer #1 (Comments to the Authors (Required)):

Summary :

In this manuscript, Grech et al. investigate the role of the Tepsin protein in actin-dependent transport from the trans-Golgi to the plant-like vacuole in *Toxoplasma gondii*. Previous studies have revealed that Apicomplexan parasites, over the years, have lost entire trafficking complexes and exhibit lower quantities of proteins involved in vesicular trafficking compared to other eukaryotes. However, the development of bioinformatic tools has identified the existence of paralogs capable of compensating for this loss. Various factors from the cytoskeleton, such as microtubules and more recently, F-actin and myosin F, have been implicated in endosomal transport. In this study, the authors use a conditional knockout TgTEP line to demonstrate its function in transport from the trans-Golgi to the vacuolar system. The knockout of Tepsin results in the loss of organization of the trans-Golgi for the cis-Golgi, and in the plant like vacuolar trafficking defect. The loss of Tepsin also induces a modification in the positioning of actin filaments and immobility of myosin F. Overall, the authors have identified a novel protein involved in

trafficking to the vacuolar compartment.

Major comments:

- After identifying AP4 ϵ through TurboID, the authors conducted IFA analysis to establish that TgTEP is a component of the ARP4 complex. This involved demonstrating the colocalization of AP4 ϵ and TgTEP, as well as examining the impact of TgTEP knockout on AP4 ϵ localization. However, a more conclusive confirmation of the existence of this interaction could be achieved by performing a co-immunoprecipitation followed by a western blot. This additional experimental approach would enhance the robustness of the findings and provide a stronger foundation for the assertion that TgTEP is indeed part of the ARP4 complex.

Minor comments :

- Figure 1B, light colors are hard to see

- Line 32; TgTEP is essential and its KO is detrimental to the TGN but not cis-Golgi section: "no change was observed for the" cis -Golgi must be missing here.

- In Figure 2, TEM was conducted on both WT and parasites 72 hours post induction. Since TgTEP is fully knockout 48h post induction, is the Golgi-apparatus intact or not? If yes, justify the exhibition of the 72h post induction results only.

- Among the set of criteria used to prioritize a list of candidates, the abundance of the protein should be included. Also, in this paper protein with a fitness score lower than -1 are considered to be essential. It is more conventional to consider as essential the protein with a fitness score of -2 and below.

- Figure 4E lacks clarity regarding whether the signal of AP4 ϵ disappears or mislocalizes 72 hours post-induction. Enhancing image quality by outlining parasites and incorporating profile plots to confirm TgTEP and AP4 ϵ colocalization would improve interpretation.

- In Figure 5A and B, consider individualizing both channels to enhance appreciation of actin filaments, which may not be clearly visible after merging.

- In figure 5D-E, it is clear that the green line indicates the tracks of the actinCb-emerald, however more explanation is needed for the red line in the lower panel of both figures. Providing additional context will improve the figure's comprehensibility.

- Could the authors provide the name of the antibodies used for the IFAs, especially for the secondary antibodies (Halo and SNAP).

Reviewer #2 (Comments to the Authors (Required)):

In the submitted paper, "Toxoplasma Tepsin is essential for actin-dependent transport from the trans-Golgi to the vacuole", the authors explore the role of an Alveolata-specific Tepsin protein in *T. gondii*. They determine that it localizes to the trans Golgi of the parasite and that conditional knockout disrupts the trans Golgi but not the anterior parasite-specific secretory organelles. They then use the conditional knockout to show that trafficking of cargo to the PLVAC is disrupted upon loss of TgTEP, and that the protein is linked to F-actin dependent transport to the PLVAC. In addition, they use proximity labelling to show TgTep likely interacts transiently with components of the micropore. Specific comments are below.

Major comments:

1. The abstract states that TgTep transiently associates with the micropore. This is an overstatement of the data. As the authors acknowledge later, any potential interaction might occur on vesicles that have been internalized (perhaps use "micropore components"). In addition, the "interaction" claimed in the results is via proximity labelling and thus may be merely proximal rather than truly interacting. The authors should adjust their claims in the appropriate areas accordingly.

2. Line 164 - The authors state that ELC, GRAs, MICs, ROPs, apicoplast, and IMC are unaffected in Fig S4. But the figure also shows that the mitochondria is fragmented (collapsing from lasso shaped to rounded spots) - this is never discussed this in the text at all.

a) Does this suggest parasite death?

b) It appears that mitochondria fragmentation occurs mostly after PLVAC disruption - this should be discussed.

c) This data should be moved to the main figures and discussed appropriately (with only unaffected organelles in the supplementary figures).

3. The authors state "To prove that is indeed interacting with TgAP4 ϵ , we performed a colocalization analysis in parasites expressing TgAP4 ϵ tagged with Halo and TgTEP-mCherry and found that both proteins are perfectly co-localised (Fig.4E)."

- Colocalization does not mean that they bind to/interact with each other.

- While colocalization plus the proximity labeling suggests interaction, interaction would be better confirmed by a co-IP.

4. For the TurboID experiments - the authors should show that the fusion protein targets perfectly (and is not present at the periphery or base of the parasite). Any mistargeting would result in spurious targets being identified via proximity labeling.

5. There are numerous issues with the TurboID or in how it is presented. These make it difficult to interpret and understand the data.

a) Supplementary Figure 6 - it is unclear what is meant by "protein hits" - is this spectral count?

b) Supplementary Table 2 needs a legend explaining what each sheet in the excel file is. It's difficult to understand which sheet they are referring to at different points in the text. It's also unclear what the "Difference" value is. Perhaps consider breaking some of these up into individual supplementary figs

c) 5 of the proteins in Supp Fig. 6D are not found in Supp. Table 2 (in any of the 4 sheets): It is unclear why these are present here and not in the Table S2

(#3) TGGT1_206580 Formin2

(#8) TGGT1_290160 SortLR

(#10) TGGT1_227800 Eps15L

(#11) TGGT1_297520 CGAR

(#14) TGGT1_235020 putative COPI protein

d) Line 254 - "Of the two proteins seen to be the most enriched, one was seen to be non-essential (TGGT1_229930). The other protein was profilin. In addition to these, K13, Eps15L, and CGAR were also enriched."

TGGT1_229930 and profilin are present in Supp Table 2, but Eps15L and CGAR are absent?

e) Line 278 - "several proteins linked to actin functions, such as FRM-2, MyoF and profilin were identified in proximity labelling"

FRM-2 is absent from Supp Table 2

f) Line 268 - Other proteins of interest identified in the datasets included trafficking-related proteins (SortLR, DrpB and clathrin heavy and light chain) that have been demonstrated to reside at the TGN."

SortLR is absent from Supp Table 2

6. The writing of the manuscript is of poor quality in many places.

Some examples

a) Line 157 - "Upon induction with rapamycin, no change was observed for the (Schmidt et al., 2023) (Pfluger et al., 2005), even after extended induction times of 72 hours (Fig. 2A)"

This sentence is incomplete?!?

b) "Having identified TgTEP as critical for material transport to the PLVAC, which receives endocytosed material (Stasic et al., 2022) and several components of the micropore, where endocytosis originates, suggests that TgTEP vesicles transiently interact with the micropore or that TgTEP plays another, more direct role in endocytosis" this needs to be rephrased for clarity

c) I don't think the abstract appropriately describes the manuscript. It seems to discuss previous work and only briefly highlights the results of the paper, often excluding important findings (e.g. the disruption of the PLVAC without ELC revealing a unique element of transport, other data from the paper).

d) Figure 7 is never referenced or discussed in the text. This model should be explained for the reader.

e) Supplementary figure 8 is never referenced in the text.

f) "MyoF, became punctate, immobile and distributed throughout the parasite (Fig. 6A-C, Movie 2)." I believe the authors should only be referring to 6A here? (B is the next sentence, C is a different experiment)

7. Line 263 - "Neither delivery of SAG1 to the plasma membrane.... Supp Fig 7 appears to show a significant decrease in de novo SAG1 and a significant increase in recycled SAG1. Am I misinterpreting something here? Please clarify for the reader.

Minor comments:

1. Line 162 - Is DrpB a TGN marker? I thought the authors previously showed it was just downstream of the Golgi? Adjust txt to be clearer? (it is called both trans Golgi and post Golgi in the manuscript)

2. Line 169- "suggesting that TgTEP is required for Golgi-organization and that its absence leads to its fragmentation and expansion." The authors appear to be indicating that the large vesicles are fragmented and then expanded Golgi vesicles? Could they also be amylopectin granules? Perhaps look by PAS staining?

3. Fig S4E - The GRA1 staining looks different at 96hrs than at 72 or control - is there enhanced secretion late?

4. Furthermore, excision of TgTEP results in mislocalisation of TgAP4ε (Fig.4E). We, therefore, conclude that TgTEP and TgAP4ε are part of the apicomplexan adaptor complex-4, which is required for the transport of material from the TGN to the PLVAC."

- TgTEP has clearly been shown for transport but for apicomplexan adaptor complex-4 - just adjust wording to clarify
- The image in 4E does not clearly show that AP-4ε mislocalizes. There's no peripheral marker so it's difficult to see where the parasites are. There appears to be similar punctae of AP-4ε staining which is what is seen in the -rapa condition too (noting the scale bar differences).

5. The videos could use a better description of what is happening (and perhaps arrows)

6. It would be helpful for resubmission if the supplementary figs were labelled as such to help the reviewers (e.g Fig S1)

Reviewer #3 (Comments to the Authors (Required)):

The authors want to investigate the localization/role of novel protein potentially a distant orthologue of Tepsin in Toxoplasma. This protein is essential for the parasite, which justifies its study. However, the main issue is that the data presented in this manuscript seem to be selected to infer a role for Tepsin in a post-Golgi transport -by analogy with human tepsin. It remains uncertain why many data are misinterpreted or overinterpreted, and some neglected as not fitting in their model. This study needs to be investigated into more than one angle to ascertain the correct localization/role of the protein. The manuscript would also benefit from a better writing with clarity for non-parasitologists and presentation of the rationale for each experiment.

The title is confusing as 'vacuole' would be confused by the parasitophorous vacuole.

Apicomplexa or apicomplexan parasites not apicomplexans

Abstract and Introduction line 75: we recently performed a phenotypic screen. A few words on how this screen has been generated, will be useful.

Abstract cites 'plant like-vacuole', 'micropore' need some physiological definitions for non-parasitologists.

PLVAC is now commonly designed by PLV

Overall the abstract is not very informative and lacks essential findings. Needs to be re-written.

Introduction:

Line 41: specify which hosts

Line 46: avoid jargon (eg huge) or throughout the manuscript (perfectly, live movies,...). Distance of Apicomplexa to other eukaryotes

Line 55: explain what are ArIX-proteins

Line 74: explain what is the apicoplast

Results

Line 117: explain what are these strains and what is a tachyzoite for non-parasitologists

Line 120-122: provide a rationale why C-terminal tagging is more trustable than HyperrLOPIT

Figure 1A: a comparison with motifs of human tepsin will be useful. In mammalian cells, tepsin comprises two phylogenetically conserved peptide motifs at the C-terminus: [GS]LFXG[ML]X[LV] and S[AV]F[SA]FLN for interaction with the C-terminal domains of the β4 and ε subunits of AP-4, respectively. However, these C-terminal motifs are absent in the Toxoplasma protein, making 'TgTEP' not a canonical tepsin. This weak homology must be discussed or the name TgTEP must be reconsidered. Figure 1B must include phylogeny with mammals.

Localization of TgTEP: the authors state that 'TgTEP resides at a well-defined subcompartment of the trans-Golgi' based on IFA. However, the signals of colocalization with endosomal-like compartment markers (eg proM2Ap,...) in Fig S1 seem more convincing than with Golgi markers. Why are the negative data in the main text and the positive data in SI? The use of BFA to induce the Go-ER collapse is not too convincing as at 100ug/ml, BFA FA can also induce endosomal compartments to form BFA-induced compartments that contain endosomal marker.

To this point, data in Fig. 3A with colocalization between CPL and TgTEP must be shown in Fig. 1 and quantified.

Figure SI3A: treatment with Rapa 72h: where are the parasites in the shown image of a big blob? Hoechst images alone will be useful. Why is the TgTEP-mCherry signal in the host cell? Same comment for Fig. 4E. If it because the PVM has ruptured, this is not relevant to look at organelle staining Fig. 2B. In images SI 4, the PV are intact. Why choosing to show images of Golgi in fragmented PV and images for other organelles in more intact PV.

With rapamycin in Fig. 2A-B, the syntaxin6 and DrpB signals seem more spread than without rapamycin, and the signal for TOM40 shows convincing images of a collapse of the mitochondrial network in Fig SI 4F. Why is this piece of information about mitochondria missing from the main text (page 7 L166)? In other words, why the subtitle is: 'TgTEP is essential and its knockout is detrimental to TGN but not cis-Golgi' and omits mitochondria.

In Fig. 2: EM show large e-lucent vesicles but what is the evidence that they are from trans-Golgi. How frequent it is? It is better to show images at low magnification, same PV size and not dividing parasites to document the absence of any Golgi in serial sections. The diffuse signal in Fig 2B by IFA for SortLR throughout the cytosol does not seem matching with these apically

located large vesicles. How can the authors rule out that these large vesicles are not PLV-derived based on IFA in Fig. 3A or the corresponding images of recycled SAG1 vesicles in Fig. S17? How do they look like the mitochondrion by EM? If mitochondrial functions are impaired, many functions will be altered and many cytopathies (organelle collapse, vesiculation, fragmentation) are expected.

What is the signal for TgPET in extracellular parasites vs. CPL and Golgi markers?

TgTEP interacts with TGGT1_244290. There is no experimental evidence that it is a Golgi AP4 protein homolog, and naming it TgAP4ε is speculative more especially that TgTEP has no motif with AP4 interaction. Would the N-terminus ENTH domain of TgTEP be involved in the interaction with an adapter-protein, this has to be demonstrated on a N-terminus truncated of TgTEP. In Fig. 4F, the signal for TgTEP and TGGT1_244290 needs to be quantified at the same exposure time as it shows almost 99% colocalization but at different contrasts. Such a level of colocalization even for 2 markers for the same organelle is unusual. The reviewer would like to see more examples for TgTEP and TGGT1_244290, on parasites from different PV size and extracellular parasites.

If TgTEP is posttail involved in the trafficking of material between PLV and the plasma membrane, how does the micropore look like under condition of TgTEP depletion?

What is the evidence that arrows in yellow in Fig. 5B are TgTEP vesicle moving towards the micropore specifically, as stated page 10 line 293. The micropore is more anterior (apical) than at mid-body.

Assays on the TgTEP-actin/myosin: What are the residues on TgTEP involved in cytoskeleton/formin interaction and why the circulation of TgTEP-vesicles is mostly confined to the anterior pole of the parasite, is not clarified. The model in Fig. 8 suggests small vesicles with TgTAP shuttling between the TGN/ELC but the IFA signal for TgTEP is very large and median -not on one side of the parasite. The confusion resides in mixing TgTEP protein and vesicles. For example, Page 11 line 337 the authors wrote 'TgTEP shows a steady state localization at the trans Golgi of the parasite near the actin nucleation centre FRM 2' where it is in the model that shows only vesicles. How can the authors rule out the association of TgTEP with the PLV compartments. More important the manuscript does not address what could be the cargoes of these TgTEP transport vesicles. The model does not integrate the link between TgTEP depletion and mitochondrion breakdown. The Discussion is more a summary of the Results section and should include these central questions.

Response to reviewers' comments for our resubmission: Tepsin and the AP4 Complex Mediate Transport from the Trans-Golgi to the Plant-Like Vacuole in Toxoplasma via a Clathrin-Dependent Pathway

We would like to thank the reviewers for their time and constructive comments that helped us to improve our manuscript.

In our revision, we provide additional data to address the reviewers' comments. Briefly, we performed additional experiments to analyse the localization of TgTEP, and its interaction with the adaptin protein AP4 ϵ and discuss the implication of TgTEP deletion for the mitochondria in more detail. To specifically address the comments of reviewer 3, we also asked Prof Joel Dacks (University of Alberta) to collaborate with us to perform an exhaustive bioinformatic and phylogenetic analysis on this protein. We also substantially edited the manuscript to clarify the reviewers' concerns.

During the revision of our manuscript a study was published in International Journal of Biological Macromolecules (He et al., 2025), which also identified Tepsin in an alternative approach as being important for transport of vesicles to the PLVAC. While the phenotype described in this study corresponds well with our characterization, our mechanistic characterization of interaction partners and the actin dependent transport is unique and novel. We are therefore convinced that our study is of significant interest to a broader readership, interested in general cell biology and vesicular transport.

Please find our detailed response below (in blue):

Reviewer #1 (Comments to the Authors (Required)):

Summary:

In this manuscript, Grech et al. investigate the role of the Tepsin protein in actin-dependent transport from the trans-Golgi to the plant-like vacuole in *Toxoplasma gondii*. Previous studies have revealed that Apicomplexan parasites, over the years, have lost entire trafficking complexes and exhibit lower quantities of proteins involved in vesicular trafficking compared to other eukaryotes. However, the development of bioinformatic tools has identified the existence of paralogs capable of compensating for this loss. Various factors from the cytoskeleton, such as microtubules and more recently, F-actin and myosin F, have been implicated in endosomal transport. In this study, the authors use a conditional knockout TgTEP line to demonstrate its function in transport from the trans-Golgi to the vacuolar system. The knockout of Tepsin results in the loss of organization of the trans-Golgi for the cis-Golgi, and in the plant like vacuolar trafficking defect. The loss of Tepsin also induces a modification in the positioning of actin filaments and immobility of myosin F. Overall, the authors have identified a novel protein involved in trafficking to the vacuolar compartment.

We would like to thank the reviewer for the time spent to assess our manuscript. We appreciate the interest in our study and the useful comments, which helped us to improve our manuscript.

Major comments:

- After identifying AP4 ϵ through TurboID, the authors conducted IFA analysis to establish that TgTEP is a component of the ARP4 complex. This involved demonstrating the colocalization of AP4 ϵ and TgTEP, as well as examining the impact of TgTEP knockout on AP4 ϵ localization. However, a more conclusive confirmation of the existence of this interaction could be achieved by performing a co-immunoprecipitation followed by a western blot. This additional experimental approach would enhance the robustness of the findings and provide a stronger foundation for the assertion that TgTEP is indeed part of the ARP4 complex.

We thank the reviewer for this comment and not only performed the suggested experiments, but also went far beyond his/her suggestion. Briefly we now include in our revision:

- Co-IP, demonstrating interaction of AP4 ϵ and TgTEP as suggested by the reviewer (see Figure 3A).
- Mass-Spec analysis of the TgTEP-IP and AP4 ϵ -IP, leading to the identification of the whole AP4-complex and clathrin (Fig.3B, C).
- Generation of a conditional mutant for AP4 ϵ and demonstration that deletion of AP4 leads to the same phenotype as deletion of TgTEP (Fig.3D, E).
- Phylogenetic analysis, demonstrating the co-evolution and conservation of TgTEP and AP4 ϵ (Fig. 1)

Minor comments:

- Figure 1B, light colors are hard to see
- Line 32; TgTEP is essential and its KO is detrimental to the TGN but not cis-Golgi section: "no change was observed for the" cis -Golgi must be missing here.

We made the relevant edits in our revision.

- In Figure 2, TEM was conducted on both WT and parasites 72 hours post induction. Since TgTEP is fully knockout 48h post induction, is the Golgi-apparatus intact or not? If yes, justify the exhibition of the 72h post induction results only.

We have now included the 48-hour post-induction (hpi) time point in our revision. At this stage, the Golgi apparatus appears to begin fragmenting, although precise assessment is challenging based on TEM imaging alone (Fig. 2E). The fragmentation reaches its peak at approximately 72 hpi, as quantified in Fig. 3F. This is why we originally focused on the 72-hour time point in our initial figure.

- Among the set of criteria used to prioritize a list of candidates, the abundance of the protein should be included. Also, in this paper protein with a fitness score lower than -1 are considered to be essential. It is more conventional to consider as essential the protein with a fitness score of -2 and below.

We have now utilized both TurboID and co-immunoprecipitation (Co-IP) approaches to prioritize our candidate proteins, as described in the manuscript. All data obtained from these assays have been deposited in ProteomeXchange (PXD057662).

Regarding the reviewer's suggestion to include protein abundance as a prioritization criterion, we opted not to use this factor, as proteins interacting with TgTEP may do so only transiently

or in a sub-stoichiometric manner while still playing a functional role. We acknowledge the reviewer's concern regarding the fitness score threshold and have revised the manuscript to clarify the criteria used for candidate selection.

"To prioritise the list of candidates that could be involved in trafficking of TgTEP-vesicles, the 30-minute timepoint dataset was analysed according to a set of criteria. Firstly, all proteins with a phenotypic score of -1 or higher were excluded from the list of potential stable interactors (Sidik et al., 2016). Next, since it was expected that stable interactors would localise to the same sub-cellular compartment as TgTEP, proteins which were observed to cluster far away from TgTEP localisation, according to hyperLOPIT (Barylyuk et al., 2020), were excluded. Additionally, proteins already described in the literature that are unlikely to directly anchor TgTEP to vesicles, such as many IMC and ribosomal proteins, were excluded." (Page 12 lines 2-10)

- Figure 4E lacks clarity regarding whether the signal of AP4ε disappears or mislocalizes 72 hours post-induction. Enhancing image quality by outlining parasites and incorporating profile plots to confirm TgTEP and AP4ε colocalization would improve interpretation.

We now included a detailed analysis of AP4 ε and a conditional mutant for AP4ε. See Figs. 2 and 3.

- In Figure 5A and B, consider individualizing both channels to enhance appreciation of actin filaments, which may not be clearly visible after merging.

We did try to depict this better. However, by individualizing the channels the trafficking of vesicles along the filaments is not obvious. We decided to keep this as it is.

- In figure 5D-E, it is clear that the green line indicates the tracks of the actin Cb-emerald, however more explanation is needed for the red line in the lower panel of both figures. Providing additional context will improve the figure's comprehensibility.

Both lines (green and red) represent tracks that were selected for quantification in the kymograph analysis. We have now included an explanation in the figure legend.

- Could the authors provide the name of the antibodies used for the IFAs, especially for the secondary antibodies (HALO and SNAP).

For HALO and SNAP no antibodies were used, but dyes that covalently bind to these tags. It is now mentioned in the manuscript and supplementary tables.

Reviewer #2 (Comments to the Authors (Required)):

In the submitted paper, "Toxoplasma Tepsin is essential for actin-dependent transport from the trans-Golgi to the vacuole", the authors explore the role of an Alveolata-specific Tepsin protein in *T. gondii*. They determine that it localizes to the trans Golgi of the parasite and that conditional knockout disrupts the trans Golgi but not the anterior parasite-specific secretory organelles. They then use the conditional knockout to show that trafficking of cargo to the

PLVAC is disrupted upon loss of TgTEP, and that the protein is linked to F-actin dependent transport to the PLVAC. In addition, they use proximity labelling to show TgTep likely interacts transiently with components of the micropore. Specific comments are below.

We would like to thank the reviewer for the time spent to assess our manuscript. We appreciate the interest in our study and the useful comments, which helped us to improve our manuscript.

Major comments:

1. The abstract states that TgTep transiently associates with the micropore. This is an overstatement of the data. As the authors acknowledge later, any potential interaction might occur on vesicles that have been internalized (perhaps use "micropore components"). In addition, the "interaction" claimed in the results is via proximity labelling and thus may be merely proximal rather than truly interacting. The authors should adjust their claims in the appropriate areas accordingly.

We agree with the reviewer and indeed this is likely the case since the depletion of TgTEP does not result in a similar phenotype compared to depletion of K13 (see (Koreny et al., 2023)). In contrast, depletion of TgTEP results in a significant accumulation of vesicles containing recycled Sag1. While we were unsure, how to interpret this result in our initial analysis, based on the reviewer's comments, we now performed a more in-depth analysis of the recycling of surface proteins, using Sag1 as a marker. This experiment is similar to the analysis performed in our original submission (Figure S11). While we did see a slight, but significant increase of recycled Sag1 (overall intensity), we did not include the analysis of vesicle size. We now performed this experiment and it is obvious that in the case of TgTEP depletion, vesicles containing recycled SAG1 are increased in size. We included this detailed analysis in our revision as new supplemental Figure 11.

We interpret these data that uptake of Sag1 through the micropore is not affected in absence of TgTEP, but that recycled material is not efficiently trafficked. Since recycling likely occurs via the PLVAC, we speculate that depletion of TgTEP leads to the interruption of recycling. In conclusion, we agree with the reviewer that interaction with components of the micropore is at best transient and additional experiments are required to elucidate this mechanism in future studies.

In our revision we focus on the role of TgTEP in the context of AP4, Clathrin and F-actin dependent transport and decided to include these new data "only" in the supplements. We changed abstract and main body of the manuscript accordingly.

2. Line 164 - The authors state that ELC, GRAs, MICs, ROPs, apicoplast, and IMC are unaffected in Fig S4. But the figure also shows that the mitochondria is fragmented (collapsing from lasso-shaped to rounded spots) - this is never discussed this in the text at all.

a) Does this suggest parasite death?

b) It appears that mitochondria fragmentation occurs mostly after PLVAC disruption - this should be discussed.

c) This data should be moved to the main figures and discussed appropriately (with only unaffected organelles in the supplementary figures).

We thank the reviewer for this suggestion. In our first submission, we decided to include these data, but to put them into the supplements, since it is somewhat distracting from the main message of the study. It is common that mitochondrial collapse can be observed in stressed or starving parasites. As a more recent example, we would like to mention the disruption of K13, a component of the micropore. Here aberrant mitochondria were also observed, likely representing a secondary phenotype (see Koreny et al., Nature Communications 2023). Furthermore, general stress or nutrient depletion can cause mitochondria to appear aberrant (Charvat and Arrizabalaga, 2016; Ovcariikova et al., 2017). Finally, it has been observed that interruption of F-actin dynamics, which is also caused by the depletion of TgTEP, can cause mitochondrial fragmentation and collapse (Shaw et al., 2000). Therefore, we do believe that the observed collapse is a secondary effect, especially since it is unlikely that a protein involved in secretory traffic has a direct influence on the mitochondria. While we decided to present this finding in the supplementary data, since it would distract from the main findings on vesicular transport and the role of TgTEP and AP4, we discussed it in more detail in our revision.

3. The authors state "To prove that is indeed interacting with TgAP4 ϵ , we performed a colocalization analysis in parasites expressing TgAP4 ϵ tagged with Halo and TgTEP-mCherry and found that both proteins are perfectly co-localised (Fig.4E)."

- Colocalization does not mean that they bind to/interact with each other.

- While colocalization plus the proximity labeling suggests interaction, interaction would be better confirmed by a co-IP.

We fully agree with the reviewer and will include these data (see also our response to reviewer 1 and Fig. 3).

4. For the TurboID experiments - the authors should show that the fusion protein targets perfectly (and is not present at the periphery or base of the parasite). Any mistargeting would result in spurious targets being identified via proximity labeling.

In our study we demonstrate that several tags do not interfere with targeting of TgTEP to Golgi derived vesicles. In Fig.3A (middle panel), we also show that after 30 mins biotinylated proteins are concentrated in the Golgi area, corresponding to the localization of TgTEP shown with other tags. However, after 6 hours of exposure to biotin, this labelling is present in more locations showing how dynamic this protein is.

Finally, we now performed standard IPs of TgTEP (new Figure 3) and identified the same proteins as in the TurboID (CHC, AP4-subunits, etc.).

5. There are numerous issues with the TurboID or in how it is presented. These make it difficult to interpret and understand the data.

a) Supplementary Figure 6 - it is unclear what is meant by "protein hits" - is this spectral count?

b) Supplementary Table 2 needs a legend explaining what each sheet in the excel file is. It's difficult to understand which sheet they are referring to at different points in the text. It's also unclear what the "Difference" value is. Perhaps consider breaking some of these up into individual supplementary figs

c) 5 of the proteins in Supp Fig. 6D are not found in Supp. Table 2 (in any of the 4 sheets): It is unclear why these are present here and not in the Table S2

- (#3) TGGT1_206580 Formin2
- (#8) TGGT1_290160 SortLR
- (#10) TGGT1_227800 Eps15L
- (#11) TGGT1_297520 CGAR
- (#14) TGGT1_235020 putative COPI protein

d) Line 254 - "Of the two proteins seen to be the most enriched, one was seen to be non-essential (TGGT1_229930). The other protein was profilin. In addition to these, K13, Eps15L, and CGAR were also enriched."

TGGT1_229930 and profilin are present in Supp Table 2, but Eps15L and CGAR are absent?

e) Line 278 - "several proteins linked to actin functions, such as FRM-2, MyoF and profilin were identified in proximity labelling" FRM-2 is absent from Supp Table 2

f) Line 268 - Other proteins of interest identified in the datasets included trafficking-related proteins (SortLR, DrpB and clathrin heavy and light chain) that have been demonstrated to reside at the TGN." SortLR is absent from Supp Table 2

The excel sheets presented were not an extensive list of all the proteins identified using TurboID. We have now added 2 more tables (Table S10 and S11) one of them including the significantly enriched proteins after 6h of incubation with biotin and another one including the proteins with a phenotypic score <-1.5 and their predicted location. The raw mass spectrometry proteomics data have been deposited to the ProteomeXchange Consortium via the PRIDE (Perez-Riverol et al., 2022) partner repository with the dataset identifier PXD057662.

We understand that these excel sheets might be confusing for the reader. At the reviewers' advice, we are willing to eliminate this confusion and remove these excel sheets, only supplying the full datasets as have been deposited to the repository.

6. The writing of the manuscript is of poor quality in many places.

Some examples

a) Line 157 - "Upon induction with rapamycin, no change was observed for the (Schmidt et al., 2023) (Pflugger et al., 2005), even after extended induction times of 72 hours (Fig. 2A)"

This sentence is incomplete?!?

We apologize since we must have erase part of the sentence before submitting the manuscript for evaluation.

b) "Having identified TgTEP as critical for material transport to the PLVAC, which receives endocytosed material (Stasic et al., 2022) and several components of the micropore, where endocytosis originates, suggests that TgTEP vesicles transiently interact with the micropore or that TgTEP plays another, more direct role in endocytosis" this needs to be rephrased for clarity

We thoroughly rewrote the manuscript and was now corrected by a native speaker.

c) I don't think the abstract appropriately describes the manuscript. It seems to discuss previous work and only briefly highlights the results of the paper, often excluding important findings (e.g. the disruption of the PLVAC without ELC revealing a unique element of transport, other data from the paper).

We thoroughly rewrote the manuscript and was now corrected by a native speaker.

d) Figure 7 is never referenced or discussed in the text. This model should be explained for the reader.

This figure is discussed in the discussion section.

e) Supplementary figure 8 is never referenced in the text.

We made sure all figures are referenced in the text

f) "MyoF, became punctate, immobile and distributed throughout the parasite (Fig. 6A-C, Movie 2)." I believe the authors should only be referring to 6A here? (B is the next sentence, C is a different experiment)

We thoroughly rewrote the manuscript and was now corrected by a native speaker.

7. Line 263 - "Neither delivery of SAG1 to the plasma membrane.... Supp Fig 7 appears to show a significant decrease in de novo SAG1 and a significant increase in recycled SAG1. Am I misinterpreting something here? Please clarify for the reader.

See our comments for reviewer 1 above. We reanalysed the data and provided additional information, as suggested.

Minor comments:

1. Line 162 - Is DrpB a TGN marker? I thought the authors previously showed it was just downstream of the Golgi? Adjust txt to be clearer? (it is called both trans Golgi and post Golgi in the manuscript)

Corrected.

2. Line 169- "suggesting that TgTEP is required for Golgi-organization and that its absence leads to its fragmentation and expansion." The authors appear to be indicating that the large vesicles are fragmented and then expanded Golgi vesicles? Could they also be amylopectin granules? Perhaps look by PAS staining?

We clarified this in our revision. A similar fragmentation has been observed for other conditional mutants for Golgi-trafficking factors such as ArlX3 (Klinger et al., 2024).

3. Fig S4E - The GRA1 staining looks different at 96hrs than at 72 or control - is there enhanced secretion late?

We did not observe enhanced secretion of dense granules. The chosen, image was selected since it demonstrates that secretion of material into the PV still occurs. GRA1 typically associates with the intravacuolar network that is organized by F-actin (see Periz et al., 2017). This can be nicely seen in the control. At 96 hours this clear accumulation is lost, likely because the network collapsed due to the observed changes in MyoF dynamics.

While it would be interesting to further investigate this behaviour, it is beyond the scope of this study, since it is at best an indirect phenotype, similar to the one observed for mitochondria.

4. Furthermore, excision of TgTEP results in mislocalisation of TgAP4ε (Fig.4E). We, therefore, conclude that TgTEP and TgAP4ε are part of the apicomplexan adaptor complex-4, which is required for the transport of material from the TGN to the PLVAC."

- TgTEP has clearly been shown for transport but for apicomplexan adaptor complex-4 - just adjust wording to clarify

- The image in 4E is does not clearly show that AP-4e mislocalizes. There's no peripheral marker so it's difficult to see where the parasites are. There appears to be similar punctae of AP-4e staining which is what is seen in the -rapa condition too (noting the scale bar differences).

We agree with the reviewer and refer to our response to reviewer 1. We included Co-IP and a phenotypic characterization for the depletion of AP4ε (Fig.3).

5. The videos could use a better description of what is happening (and perhaps arrows)
Corrected.

6. It would be helpful for resubmission if the supplementary figs were labelled as such to help the reviewers (e.g Fig S1)
Corrected.

Reviewer #3 (Comments to the Authors (Required)):

The authors want to investigate the localization/role of novel protein potentially a distant orthologue of Tepsin in *Toxoplasma*. This protein is essential for the parasite, which justifies its study. However, the main issue is that the data presented in this manuscript seem to be selected to infer a role for Tepsin in a post-Golgi transport -by analogy with human tepsin. It remains uncertain why many data are misinterpreted or overinterpreted, and some neglected as not fitting in their model. This study needs to be investigated into more than one angle to ascertain the correct localization/role of the protein. The manuscript would also benefit from a better writing with clarity for non-parasitologists and presentation of the rationale for each experiment.

We would like to thank the reviewer for the time spent to assess our manuscript. We appreciate the interest in our study and find some of the comments useful. We are however unsure in which way the reviewer finds the results of our study biased.

First of all, this reviewer seems to believe that we simply performed a homology search on Tepsins and used findings from human tepsins to "simply" reproduce them in *Toxoplasma gondii*. Nothing could be further from reality, and we did NOT select data to infer anything nor were we biased in data interpretation.

First, this protein was identified (as mentioned in the manuscript) in a complex phenotypic screen, aimed to identify proteins that are required for F-actin dynamic dependent processes (see Li et al., Nature Microbiology 2022 (Li et al., 2022)) and as clearly stated in the abstract:

"We previously performed a phenotypic screen to determine the function of essential genes, only found in apicomplexan parasites and identified a hypothetical protein (TGGT1_301410). "

While our initial bioinformatic analysis suggested the presence of an ENTH domain, we didn't believe that this protein is the homologue of Tepsin and indeed it was listed as "hypothetical protein" in ToxoDB. Instead, we performed our analysis unbiased and found:

- Localization to the trans-Golgi
- Close proximity and potential interaction with AP4e

It was then, that we realized that it could be a distant orthologue of Tepsin and performed more in-depth phylogenetic analysis. However, we do agree with the reviewer that the overview presented in figure 1 in our first submission can be optimized and presented in a more informative way. We therefore teamed up with the Dacks lab (University of Alberta), who are experts in the evolution of the secretory pathway in eukaryotes and generated an extensive phylogenetic analysis on Tepsins and AP-complexes, as now presented in Figure 1 and several supplemental figures in our revision (see also response to reviewer 1 and 2).

Many comments of reviewer 3 refer mainly to our writing style and not to our experimental approach or data interpretation. Please find our detailed response below.

The title is confusing as 'vacuole' would be confused by the parasitophorous vacuole. We agree and will change to Plant-like vacuole (PLVAC).

Apicomplexa or apicomplexan parasites not apicomplexans

We changed to "Apicomplexa or apicomplexan parasites" where appropriate, though "apicomplexans" is broadly used in the field.

Abstract and Introduction line 75: we recently performed a phenotypic screen. A few words on how this screen has been generated, will be useful.

Based on our new data, we decided to rewrite the abstract and hope it is now more fitting.

Abstract cites 'plant like-vacuole', 'micropore' need some physiological definitions for non-parasitologists.

We included a short description in the introduction section.

PLVAC is now commonly designed by PLV

Both research teams, the Moreno and Carruthers lab, who identified this organelle in *T.gondii* suggested to name it PLVAC and not VAC (given name by the Carruthers lab) or PLV (given name by the Moreno lab), see Stasic et al., 2022

Overall the abstract is not very informative and lacks essential findings. Needs to be rewritten.

The abstract has been substantially rewritten and corrected by a native speaker.

Introduction:

Line 41: specify which hosts

We are not sure what the reviewer refers to. As we mention in the introduction apicomplexans infect human and animals.

Line 46: avoid jargon (eg huge) or throughout the manuscript (perfectly, live movies,...). Distance of Apicomplexa to other eukaryotes

Line 55: explain what are ArlX-proteins
Line 74: explain what is the apicoplast
Corrected.

Results

Line 117: explain what are these strains and what is a tachyzoite for non-parasitologists
Corrected (See page 6 lines 20-21).

Line 120-122: provide a rationale why C-terminal tagging is more trustable than HyperrLOPIT
This whole section was rewritten to insert the new phylogenetic analysis we performed.
However, we included a rationale to why we chose to perform C-terminal tagging in our proteins of interest in page 8 lines 13-14.

“To enable live imaging and localisation analysis of TgTEP, the gene was tagged with fluorescent or self-labelling tags (Supp. Fig. 3A-C). All versions of endogenously tagged TgTEP localised identically to the Golgi body, with some signal also observed throughout the cytoplasm, indicating a dynamic protein (Fig. 2A-D; Supp. Fig. 3C).”

Figure 1A: a comparison with motifs of human terpin will be useful. In mammalian cells, tepsin comprises two phylogenetically conserved peptide motifs at the C-terminus: [GS]LFXG[ML]X[LV] and S[AV]F[SA]FLN for interaction with the C-terminal domains of the β 4 and ϵ subunits of AP-4, respectively. However, these C-terminal motifs are absent in the Toxoplasma protein, making 'TgTEP' not a canonical tepsin. This weak homology must be discussed or the name TgTEP must be reconsidered. Figure 1B must include phylogeny with mammals.

We teamed up with the Dacks lab to provide an extensive phylogenomic analysis, including an analysis of the AP4-interacting domains (see Fig. 1 and Supp. Fig. 1, 2 and 12).

Localization of TgTEP: the authors state that 'TgTEP resides at a well-defined subcompartment of the trans-Golgi' based on IFA. However, the signals of colocalization with endosomal-like compartment markers (eg proM2Ap,...) in Fig S1 seem more convincing than with Golgi markers. Why are the negative data in the main text and the positive data in SI?

As described in the original publication by the Carruthers lab (see Harper et al., 2006), the marker proM2AP localizes to the **trans-Golgi network and early endosomes** (see Fig.1E from Harper et al., 2006):

Figure 1E, Harper et al., 2006:
The authors demonstrate that an antibody against the pro-peptide of MIC2, proM2AP stains both, the TGN and early endosomes.

This means that a TGN-protein like TgTEP is supposed to co-localise with proM2AP, as mentioned in our manuscript. However, only co-localisation with the TGN area, but not early

endosomes was obvious (see (now included) arrows for non-co-localisation between TgTEP and proM2AP; Supp. Fig. 3H):

To address the reviewers concerns in more detail: We showed several Golgi- and TGN markers and since proM2AP stains both, TGN and early endosomes it was included in this analysis. Importantly, as demonstrated by live imaging, TgTEP is a dynamic protein that is accumulated at the TGN, but also stains additional vesicles. According to our interpretation, it only co-localises with proM2AP at the TGN area, but not elsewhere. We hope this convinces the reviewer regarding the location of TgTep.

The use of BFA to induce the Go-ER collapse is not too convincing as at 100ug/ml, BFA FA can also induce endosomal compartments to form BFA-induced compartments that contain endosomal marker.

The BFA experiment has been performed according to the protocols provided in previous studies. Together with the data provided in Figure 3 (now Supp. Fig. 5A) regarding Golgi-collapse and the EM analysis, we strongly believe that the BFA treatment does result in redistribution of TgTEP (and as control SORTLR), demonstrating an association with the Golgi.

To this point, data in Fig. 3A with colocalization between CPL and TgTEP must be shown in Fig. 1 and quantified.

While we appreciate that the reviewer has a different preference for data presentation, we believe that the current order of figures follows a logical rationale and is well suited to the narrative of the manuscript. All data have been quantified, and to accommodate the newly provided data, we have made many adjustments to the figures.

Figure SI3A: treatment with Rapa 72h: where are the parasites in the shown image of a big blob? Hoechst images alone will be useful. Why is the TgTEP-mCherry signal in the host cell? The reviewer mistakes the signal for TgTEP-mCherry as background signal that is sometimes observed in host cells. As shown in other Figures, 72h post induction no TgTEP is detectable (see for example Fig.2F,G, Fig.3D, Fig.5A,C,D, Fig.S6A, C, E and F). In our experience it is not unusual that host cells show some background fluorescence in some channels.

Same comment for Fig. 4E. If it because the PVM has ruptured, this is not relevant to look at organelle staining Fig. 2B. In images SI 4, the PV are intact. Why choosing to show images of Golgi in fragmented PV and images for other organelles in more intact PV.

As stated above, the signal observed in Fig. 4E is background resulting from antibody amplification or natural host cell autofluorescence in that channel, rather than PVM rupture. In Supplementary Figure 4, all PVs are intact. The images presented were selected to best

illustrate the phenotypes observed for each organelle, which may naturally vary in appearance due to their respective staining conditions and biological context.

With rapamycin in Fig. 2A-B, the syntaxin6 and DrpB signals seem more spread than without rapamycin, and the signal for TOM40 shows convincing images of a collapse of the mitochondrial network in Fig SI 4F. Why is this piece of information about mitochondria missing from the main text (page 7 L166)? In other words, why the subtitle is: 'TgTEP is essential and its knockout is detrimental to TGN but not cis-Golgi' and omits mitochondria. We are not exactly sure what this reviewer refers to. In Figure 2A,B we compare the localization of the cis-Golgi marker GRASP and the trans-Golgi marker SORTLR in dependence of TgTEP expression. In absence of rapamycin TgTEP is present and in rapamycin presence, we knock out TgTEP, meaning that expression gets lost. In absence of rapamycin, both markers localize to the cis- and trans-Golgi respectively. In contrast, upon deletion of TgTEP, only the trans-Golgi (SORTLR) shows disruption, whereas the localization of GRASP remains normal. We show in Fig.S3A-D other markers that are localized to the TGN (Syntaxin 6 and DrpB) and TGN/early endosomes (proM2AP) and observe (as expected) relocation of the trans-Golgi, but not early endosomal marker.

Regarding the collapse of the mitochondria, see our response to reviewer 2 above.

In Fig. 2: EM show large e-lucent vesicles but what is the evidence that they are from trans-Golgi. How frequent it is? It is better to show images at low magnification, same PV size and not dividing parasites to document the absence of any Golgi in serial sections. The diffuse signal in Fig 2B by IFA for SortLR throughout the cytosol does not seem matching with these apically located large vesicles. How can the authors rule out that these large vesicles are not PLV-derived based on IFA in Fig. 3A or the corresponding images of recycled SAG1 vesicles in Fig. SI7? How look like the mitochondrion by EM? If mitochondrial functions are impaired, many functions will be altered and many cytopathies (organelle collapse, vesiculation, fragmentation) are expected.

We thank the reviewer for this comment and acknowledge the challenges in interpreting TEM images. We agree that we cannot definitively rule out that these vesicles are PLVAC-derived, and we do not explicitly state that they originate from the Golgi. However, the key finding from this experiment is the complete absence of an intact Golgi stack in TEM images. Whether the large electron-lucent structures are Golgi-derived, PLVAC-derived, SAG1-positive vesicles, or a combination of these is beyond the scope of this study.

To clarify this point, we will revise the text as follows:

"Next, we performed electron microscopy to analyze the ultrastructure of the Golgi. Consistent with the results presented above, no intact Golgi apparatus can be observed 72 hours post-induction. Instead, large electron-lucent vesicles are formed. While their precise origin remains uncertain, these enlarged vesicles could be derived from the Golgi, the expanded PLVAC (Fig. 3), or both. Together, these findings suggest that TgTEP is required for Golgi organization, and its absence leads to Golgi fragmentation and expansion (Fig. 2D-E)."

What is the signal for TgPET in extracellular parasites vs. CPL and Golgi markers?

We have examined CPL localization in extracellular parasites, as there is no evidence suggesting a differential organization of the Golgi between intra- and extracellular parasites.

To clarify this, we have now included an additional image and a signal distribution plot in *Supplementary Figure 3J, K*. Our data show that CPL and TgTEP do not co-localize in extracellular parasites, but TgTEP is predominantly localized at the trans-Golgi network (TGN). This observation contrasts with the recent findings by He et al. (2025) but is consistent with the localization reported by Pasquarelli et al. (2024).

TgTEP interacts with TGGT1_244290. There is no experimental evidence that it is a Golgi AP4 protein homolog, and naming it TgAP4ε is speculative more especially that TgTEP has no motif with AP4 interaction. Would the N-terminus ENTH domain of TgTEP be involved in the interaction with an adapter-protein, this has to be demonstrated on a N-terminus truncated of TgTEP.

This concern demonstrates some lack of knowledge regarding the research on trafficking factors in apicomplexans. We would like to refer the reviewer to the following studies from the Dacks laboratory regarding the evolution of adaptin complexes in diverse eukaryotic phyla, including apicomplexans:

- 1: Hirst J, Barlow LD, Francisco GC, Sahlender DA, Seaman MN, Dacks JB, Robinson MS. The fifth adaptor protein complex. *PLoS Biol.* 2011 Oct;9(10):e1001170. doi: 10.1371/journal.pbio.1001170. Epub 2011 Oct 11. Erratum in: *PLoS Biol.* 2012 Mar;10(3). doi: 10.1371/annotation/89dff893-c156-44bb-a731-bfcc91843583. PMID: 22022230; PMCID: PMC3191125.
- 2: Maciejowski WJ, Gile GH, Jerlström-Hultqvist J, Dacks JB. Ancient and pervasive expansion of adaptin-related vesicle coat machinery across Parabasalia. *Int J Parasitol.* 2023 Apr;53(4):233-245. doi: 10.1016/j.ijpara.2023.01.002. Epub 2023 Mar 8. PMID: 36898426.
- 3: Richardson E, Dacks JB. Distribution of membrane trafficking system components across ciliate diversity highlights heterogeneous organelle-associated machinery. *Traffic.* 2022 Apr;23(4):208-220. doi: 10.1111/tra.12834. Epub 2022 Mar 1. PMID: 35128766.
- 4: Nevin WD, Dacks JB. Repeated secondary loss of adaptin complex genes in the Apicomplexa. *Parasitol Int.* 2009 Mar;58(1):86-94. doi: 10.1016/j.parint.2008.12.002. Epub 2008 Dec 24. PMID: 19146987.
- 5: Dacks JB, Poon PP, Field MC. Phylogeny of endocytic components yields insight into the process of nonendosymbiotic organelle evolution. *Proc Natl Acad Sci U S A.* 2008 Jan 15;105(2):588-93. doi: 10.1073/pnas.0707318105. Epub 2008 Jan 8. PMID: 18182495; PMCID: PMC2206580.

In our revision we now included an analysis of AP4, which not only directly interacts with TgTEP (Fig.3), but also shows the same phenotype when deleted.

Regarding the truncation experiments, we do agree that this will be useful in subsequent studies, but is at this stage beyond the scope of this study.

In Fig. 4F, the signal for TgTEP and TGGT1_244290 needs to be quantified at the same exposure time as it shows almost 99% colocalization but at different contrasts. Such a level of colocalization even for 2 markers for the same organelle is unusual. The reviewer would like to see more examples for TgTEP and TGGT1_244290, on parasites from different PV size and extracellular parasites.

We appreciate the reviewer's suggestion to quantify the colocalization of TgTEP and TGGT1_244290. While we have not performed Pearson correlation analysis for this specific colocalization, we have independently validated the interaction between these two proteins through CoIP and mass spectrometry (now included in ProteomeXchange PXD057662). These biochemical approaches provide strong evidence for their association.

Regarding the imaging conditions, we acknowledge that contrast differences can affect the perceived colocalization. However, we ensured that all figures display comparable exposures levels.

If TgTEP is potentially involved in the trafficking of material between PLV and the plasma membrane, how does the micropore look like under condition of TgTEP depletion?
We did not analyse the micropore upon depletion of TgTEP. Since uptake of SAG1 is normal (Fig.S11), we do not expect any change of the micropore itself (see also response to reviewer 1 and 2).

What is the evidence that arrows in yellow in Fig. 5B are TgTEP vesicle moving towards the micropore specifically, as stated page 10 line 293. The micropore is more anterior (apical) than at mid-body.

We state that:

*“These vesicles were not only observed to move along actin filaments around the Golgi region (Fig. 5A, Movie 1), but also **along the periphery of the parasites**, consistent with TgTEP’s transient proximity to the micropore (Fig.5B).”*

The micropore is not as apical as the reviewer seems to believe, we would like to provide this image of the micropore, taken from a recent study by Koreny et al., 2023:

As can be seen in the time lapse movie and selected still images in Fig.4, these vesicles are not static but move ALONG the periphery. Some are exactly, where the micropore is localized. Please compare the stills below with the images of the micropore above.

While we agree that this hypothesis can be seen sceptical. In combination with the proximity labelling data, we do speculate that these results might indicate a transient interaction, but

we only state that **it is consistent with this hypothesis and future studies will certainly help to further elucidate this conundrum.**

Assays on the TgTEP-actin/myosin: What are the residues on TgTEP involved in cytoskeleton/formin interaction and why the circulation of TgTEP-vesicles is mostly confined to the anterior pole of the parasite, is not clarified.

We thank the reviewer for this insightful suggestion. Identifying the specific residues in TgTEP involved in cytoskeleton and formin interactions is indeed an important question. However, this detailed biochemical characterization is beyond the scope of the present manuscript and will be addressed in future studies. Given the size of TgTEP (1033 amino acids), defining the precise binding regions remains a complex task that requires extensive structural and biochemical investigations.

For the reviewer's information, our future research will include biochemical studies to purify TgTEP-positive vesicles and further explore the mechanistic role of AP4 complexes in trafficking. However, these experiments extend beyond the scope of the current study.

What we do establish for the first time in this manuscript is that TgTEP- and AP4-dependent vesicular transport is directly linked to F-actin dynamics. Regarding the confinement of TgTEP vesicles to the anterior pole of the parasite, this localization is likely due to the fact that TgTEP is required for vesicle trafficking between the trans-Golgi network and the PLVAC, both of which are positioned at the anterior pole. While F-actin and motor proteins such as MyoF are distributed throughout the cytosol and contribute to multiple trafficking processes, their involvement in TgTEP-independent pathways is expected, given their broader roles in intracellular transport.

The model in Fig. 8 suggests small vesicles with TgTAP shuttling between the TGN/ELC but the IFA signal for TgTEP is very large and median -not on one side of the parasite. The confusion resides in mixing TgTEP protein and vesicles. For example, Page 11 line 337 the authors wrote 'TgTEP shows a steady state localization at the trans Golgi of the parasite near the actin nucleation centre FRM 2' where is it in the model that shows only vesicles. How the authors can rule out the association of TgTEP with the PLV compartments. More important the manuscript does not address what could be the cargoes of these TgTEP transport vesicles. The model does not integrate the link between TgTEP depletion and mitochondrion breakdown. The Discussion is more a summary of the Results section and should include these central questions.

We are slightly confused by this assessment. We believe TgTEP behaves like most trafficking factors. For example, Rab-GTPases are accumulated usually at the donor or acceptor compartment, where we can usually see a strong IFA-signals. Yet, no one would doubt that Rab-GTPases are required for the transport of small vesicles from a donor to an acceptor compartment. Same for SNAREs, adaptins, etc.

Concerning the cargo of TgTEP-positive vesicles, our data demonstrate that TgTEP is required for protein transport to the PLVAC. This is supported by our findings with CPL and the

identification of additional PLVAC residents such as the channel CRT in Co-IP assays. We explicitly state in both the manuscript and model:

“The proposed model shows TgTEP bound to vesicles via AP-4. These vesicles are transported from the TGN to the PLVAC (orange arrow). Some vesicles might also fuse with vesicles derived via endocytosis from the micropore and subsequently travel to the PLVAC (red arrow). Knockout of TgTEP results in the retention of vesicles within the trans-Golgi network resulting in its vesiculation, and possibly a downstream inability for the parasite to metabolise endocytosed nutrients. Despite the structure of the Golgi compartment being compromised, the other secretory pathways (some of which are denoted by the grey arrows) are unaffected.”

In case the reviewer asks us to isolate the vesicles, perform mass spec analysis to identify all cargo molecules, it is again beyond the scope of this study, but will be addressed in follow up studies.

- Charvat, R.A., and G. Arrizabalaga. 2016. Oxidative stress generated during monensin treatment contributes to altered *Toxoplasma gondii* mitochondrial function. *Scientific reports*. 6:22997.
- Klinger, C.M., E. Jimenez-Ruiz, T. Mourier, A. Klingl, L. Lemgruber, A. Pain, J.B. Dacks, and M. Meissner. 2024. Evolutionary analysis identifies a Golgi pathway and correlates lineage-specific factors with endomembrane organelle emergence in apicomplexans. *Cell Rep*. 43:113740.
- Koreny, L., B.N. Mercado-Saavedra, C.M. Klinger, K. Barylyuk, S. Butterworth, J. Hirst, Y. Rivera-Cuevas, N.R. Zaccai, V.J.C. Holzer, A. Klingl, J.B. Dacks, V.B. Carruthers, M.S. Robinson, S. Gras, and R.F. Waller. 2023. Stable endocytic structures navigate the complex pellicle of apicomplexan parasites. *Nat Commun*. 14:2167.
- Li, W., J. Grech, J.F. Stortz, M. Gow, J. Periz, M. Meissner, and E. Jimenez-Ruiz. 2022. A splitCas9 phenotypic screen in *Toxoplasma gondii* identifies proteins involved in host cell egress and invasion. *Nat Microbiol*. 7:882-895.
- Ovciarikova, J., L. Lemgruber, K.L. Stilger, W.J. Sullivan, and L. Sheiner. 2017. Mitochondrial behaviour throughout the lytic cycle of *Toxoplasma gondii*. *Scientific reports*. 7:42746.
- Shaw, M.K., H.L. Compton, D.S. Roos, and L.G. Tilney. 2000. Microtubules, but not actin filaments, drive daughter cell budding and cell division in *Toxoplasma gondii*. *J Cell Sci*. 113 (Pt 7):1241-1254.

April 21, 2025

RE: JCB Manuscript #202312109R-A

Markus Meissner
Ludwig-Maximilians-University Munich

Dear Prof. Meissner:

Thank you for submitting your revised manuscript entitled "Tepsin and AP4 Mediate Transport from the trans-Golgi to the Plant-Like Vacuole in Toxoplasma". You will see that the reviewers are now overall supportive of publication. Therefore, we would be happy to publish your paper in JCB pending final revisions necessary to meet our formatting guidelines (see details below) along with addressing the final minor reviewer comments. Specifically, along with addressing the text edits please either include the requested western if feasible or temper the claims about degradation unless you are able to provide alternative supporting evidence such as a quantification of total fluorescence.

A. MANUSCRIPT ORGANIZATION AND FORMATTING:

- 1) Text limits: Character count for Articles is < 40,000, not including spaces. Count includes abstract, introduction, results, discussion, and acknowledgments. Count does not include title page, figure legends, materials and methods, references, tables, or supplemental legends.
- 2) Figures limits: Articles may have up to 10 main text figures.
- 3) * Figure formatting: Scale bars must be present on all microscopy images, including inset magnifications (you may alternatively indicate the diameter of the inset). Molecular weight or nucleic acid size markers must be included on all gel electrophoresis. Aspect ratios of images may not be altered.
- 4) Statistical analysis: Error bars on graphic representations of numerical data must be clearly described in the figure legend. The number of independent data points (n) represented in a graph must be indicated in the legend. Statistical methods should be explained in full in the materials and methods. For figures presenting pooled data the statistical measure should be defined in the figure legends. Please also be sure to indicate the statistical tests used in each of your experiments (either in the figure legend itself or in a separate methods section) as well as the parameters of the test (for example, if you ran a t-test, please indicate if it was one- or two-sided, etc.). Also, if you used parametric tests, please indicate if the data distribution was tested for normality (and if so, how). If not, you must state something to the effect that "Data distribution was assumed to be normal but this was not formally tested."
- 5) Abstract and title: The abstract should be no longer than 160 words and should communicate the significance of the paper for a general audience. The title should be less than 100 characters including spaces. Make the title concise but accessible to a general readership.
- 6) Materials and methods: Should be comprehensive and not simply reference a previous publication for details on how an experiment was performed. Please provide full descriptions in the text for readers who may not have access to referenced manuscripts.
- 7) All antibodies, cell lines, animals, and tools used in the manuscript should be described in full, including accession numbers for materials available in a public repository such as the Resource Identification Portal. Please be sure to provide the sequences for all of your primers/oligos and RNAi constructs in the materials and methods. You must also indicate in the methods the source, species, and catalog numbers (where appropriate) for all of your antibodies. Please also indicate the acquisition and quantification methods for immunoblotting/western blots.
- 8) Microscope image acquisition: The following information must be provided about the acquisition and processing of images:
 - a. Make and model of microscope
 - b. Type, magnification, and numerical aperture of the objective lenses
 - c. Temperature
 - d. Imaging medium
 - e. Fluorochromes

- f. Camera make and model
- g. Acquisition software
- h. Any software used for image processing subsequent to data acquisition. Please include details and types of operations involved (e.g., type of deconvolution, 3D reconstitutions, surface or volume rendering, gamma adjustments, etc.).

10) * Supplemental materials: There are strict limits on the allowable amount of supplemental data. Articles may have up to 5 supplemental figures. When reducing your SI count please be sure to correct the callouts in the text to reflect all changes. Please also note that tables, like figures, should be provided as individual, editable files. A summary of all supplemental material should appear at the end of the Materials and methods section.

13) ORCID IDs: ORCID IDs are unique identifiers allowing researchers to create a record of their various scholarly contributions in a single place. Please note that ORCID IDs are now *required* for all authors. At resubmission of your final files, please be sure to provide your ORCID ID and those of all co-authors.

Please note that JCB now requires authors to submit Source Data used to generate figures containing gels and Western blots with all revised manuscripts. This Source Data consists of fully uncropped and unprocessed images for each gel/blot displayed in the main and supplemental figures. For assays performed using capillary electrophoresis and/or immunoassay-based detection, authors should instead provide the electropherogram graph(s) for each experiment, plotting fluorescence/chemiluminescence intensity vs. molecular weight/size. Please be sure to provide one Source Data file for each figure gels, blots, and/or capillary electrophoresis assays along with your revised manuscript files. File names for Source Data figures should be alphanumeric without any spaces or special characters (i.e., SourceDataF#, where F# refers to the associated main figure number or SourceDataFS# for those associated with Supplementary figures). For traditional gels and blots, the lanes of the gels/blots should be labeled as they are in the associated figure, the place where cropping was applied should be marked (with a box), and molecular weight/size standards should be labeled wherever possible. For capillary electrophoresis assays, each trace in the graph should be color-coded and labeled to indicate which protein, gene, or sample is being measured (please try to avoid red/green combinations to accommodate our color-blind readers).

Journal of Cell Biology now requires a data availability statement for all research article submissions. These statements will be published in the article directly above the Acknowledgments. The statement should address all data underlying the research presented in the manuscript. Please visit the JCB instructions for authors for guidelines and examples of statements at (<https://rupress.org/jcb/pages/editorial-policies#data-availability-statement>).

B. FINAL FILES:

Thank you for your attention to these final processing requirements. Please revise and format the manuscript and upload materials within 7 days. If you need an extension for whatever reason, please let us know and we can work with you to determine a suitable revision period.

Thank you for this interesting contribution, we look forward to publishing your paper in Journal of Cell Biology.

Sincerely,

Elizabeth Miller, PhD
Monitoring Editor

Andrea L. Marat, PhD
Deputy Editor

Journal of Cell Biology

Reviewer #1 (Comments to the Authors (Required)):

The authors have been very responsive to the reviews. Especially the new comprehensive phylogeny of the players (and beyond) involved expands the scope of the study.

Reviewer #2 (Comments to the Authors (Required)):

The authors have addressed most of the points from the first revision - minor changes are listed below

1. The authors randomly switch between capitalizing and not-capitalizing "apicomplexans". My understanding is that "Apicomplexa" is capitalized as it is the phylum and "apicomplexans" is not. Please at least be consistent.
2. The authors removed the previous Fig 1A with the diagram of the protein showing the ENTH domain and disordered regions. This takes little room and would still fit well with the existing text. I recommend including this in the revision
3. new Supplementary fig 2. What is tENTH?
4. p10, L23 - the authors state - "Furthermore, excision of TgTEP caused mislocalization and degradation of TgAP4ε within 72hr after induction (Fig. 2G), suggesting a direct interaction of both proteins."
The degradation should be backed up by western blot as it is possible that the protein is at the same level but merely dispersed in the cytoplasm and thus difficult to detect by immunofluorescence.
5. p14/15 - "the clustering of MyoF may result from an excess of this protein in the absence of TgTEP". A western blot would be really helpful for this point (and easy to do).

We thank the reviewers for their constructive feedback and for acknowledging the improvements made in the revised manuscript.

Reviewer #1 (Comments to the Authors (Required)):

The authors have been very responsive to the reviews. Especially the new comprehensive phylogeny of the players (and beyond) involved expands the scope of the study.

We thank the reviewer for the positive assessment of our manuscript.

Reviewer #2 (Comments to the Authors (Required)):

The authors have addressed most of the points from the first revision - minor changes are listed below

1. The authors randomly switch between capitalizing and not-capitalizing "apicomplexans". My understanding is that "Apicomplexa" is capitalized as it is the phylum and "apicomplexans" is not. Please at least be consistent.

We appreciate the clarification and have now ensured consistency throughout the manuscript.

2. The authors removed the previous Fig 1A with the diagram of the protein showing the ENTH domain and disordered regions. This takes little room and would still fit well with the existing text. I recommend including this in the revision

We thank the reviewer for this suggestion and included previous Fig1A.

3. new Supplementary fig 2. What is tENTH?

tENTH stands for *Toxoplasma* ENTH. We include now this clarification in the figure legend.

4. p10, L23 - the authors state - "Furthermore, excision of TgTEP caused mislocalization and degradation of TgAP4 ϵ within 72hr after induction (Fig. 2G), suggesting a direct interaction of both proteins."

The degradation should be backed up by western blot as it is possible that the protein is at the same level but merely dispersed in the cytoplasm and thus difficult to detect by immunofluorescence.

We toned down this interpretation in our revision.

5. p14/15 - "the clustering of MyoF may result from an excess of this protein in the absence of TgTEP". A western blot would be really helpful for this point (and easy to do).

We apologise that this sentence has been misunderstood by the reviewer. We do not believe that in total more MyoF is present in absence of TgTEP. Our interpretation is that the same amount of MyoF is present, but that interaction partners are missing (TgTEP/AP4) resulting in clustering of those MyoF-molecules normally interacting with TgTEP/AP4.

We clarified this in the revision and now state that:

"These results suggest that the MyoF population, which normally interacts with TgTEP forms these clusters in the absence of TgTEP. However, this effect does not appear to be essential for other MyoF-mediated transport pathways, such as dense granule motility."